 **eLIFE**

# High-resolution mapping of heteroduplex DNA formed during UV-induced and spontaneous mitotic recombination events in yeast

Yi Yin[†], Margaret Dominska, Eunice Yim, Thomas D Petes*

Department of Molecular Genetics and Microbiology and University Program in Genetics and Genomics, Duke University Medical Center, Durham, United States

**Abstract** In yeast, DNA breaks are usually repaired by homologous recombination (HR). An early step for HR pathways is formation of a heteroduplex, in which a single-strand from the broken DNA molecule pairs with a strand derived from an intact DNA molecule. If the two strands of DNA are not identical, there will be mismatches within the heteroduplex DNA (hetDNA). In wild-type strains, these mismatches are repaired by the mismatch repair (MMR) system, producing a gene conversion event. In strains lacking MMR, the mismatches persist. Most previous studies involving hetDNA formed during mitotic recombination were restricted to one locus. Below, we present a global mapping of hetDNA formed in the MMR-defective *mlh1* strain. We find that many recombination events are associated with repair of double-stranded DNA gaps and/or involve Mlh1-independent mismatch repair. Many of our events are not explicable by the simplest form of the double-strand break repair model of recombination.

*For correspondence: tom. petes@duke.edu

Present address: [†]Department of Genome Sciences, University of Washington, Seattle, United States

Competing interests: The authors declare that no competing interests exist.

## Introduction

Homologous recombination (HR) is important for repairing double-stranded DNA breaks (DSBs) in diploid yeast. Although several different HR pathways have been described (*Symington et al., 2014*), the earliest steps of all pathways have a common intermediate in which a single DNA strand from one duplex invades the homologous template (green boxed region of *Figure 1*). In *Figure 1*, chromosomes are shown as double-stranded DNA structures with the two homologs drawn in different colors. Following the DSB, the broken ends are processed by 5' to 3' degradation in a two-step process with limited resection (about 100 bases) performed by Sae2 and the Mre11-Rad50-Xrs2 proteins, followed by more extensive resection (>1 kb) utilizing the redundant pathways of Exo1 or Dna2 with the Sgs1-Top3-Rmi1 complex (*Symington, 2014*). The 3' single strand derived from the broken end then invades the unbroken template chromosome forming a heteroduplex, a region of duplex DNA derived from the different DNA molecules. In *Figure 1*, heteroduplexes are shown as paired strands of different colors.

For all pathways, the invading 3' end is used as a primer to catalyze DNA synthesis (shown as red dotted lines). The subsequent steps differ for each pathway. In the one-ended synthesisdependent strand-annealing (SDSA) pathway (*Figure 1A*), following DNA synthesis, the invading strand is displaced and reanneals to the other processed broken end. Repair of the single-stranded gap (dotted blue line) completes the event. In this pathway, a region of heteroduplex occurs on only one of the two homologs, and SDSA events are not associated with crossovers.

In the two-ended SDSA pathway (*Figure 1B*), DNA synthesis is primed from both broken ends. Before ligation of the DNA strands to form a double Holliday junction, the invading end dissociates

**Figure 1.** The DSBR repair pathways. The chromatids are shown as double-stranded molecules with arrows at the 3' ends. The broken chromatid is colored blue and the intact template chromatid is red. Dotted lines indicate newly-synthesized DNA. Regions of heteroduplex DNA (hetDNA) are outlined by black boxes. All pathways are initiated by invasion of one processed broken end into the unbroken chromatid, forming a D-loop. (**A**) One-ended Synthesis-dependent strand annealing (SDSA). Following DNA synthesis primed by the invading 3' end, the invading end dissociates from the intact molecule, and re-anneals with the second broken end. The net result is an NCO event with hetDNA extending uni-directionally from the break. (**B**) Two-ended SDSA. In this mechanism, following DNA synthesis from the invading end, the other broken end pairs with the D-loop and initiates DNA synthesis. Before ligation to generate a dHJ intermediate, the invading end disengages from the template and re-pairs with the other broken end. The net result is a single chromatid with a bi-directional heteroduplex that has a strand switch at the site of the initiating DSB. (**C**) Double Holliday junction (dHJ) intermediate. Following DNA synthesis from one invading end, the second broken end pairs with the D-loop (second-end capture). This intermediate can be dissolved by migrating the two junctions inwards, followed by decatenation; this pathway results in a NCO with a bi-directional heteroduplex, identical to the product of the two-ended SDSA mechanism. Alternatively, it can also be resolved by cutting the junctions symmetrically to generate NCOs (cuts at arrows marked 1, 2, 3, and 4 or 5, 6, 7, and 8), or asymmetrically (cuts at arrows marked 2, 4, 5, and 6 or 1, 3, 7 and 8) to generate COs. For both types of resolution, the heteroduplexes are located on different chromatids, flanking the DSB site, and pointing in opposite directions. Note that cuts at positions 1, 3, 7, and eight reflect nick-directed resolution of the junctions. (**D**) Break-induced replication (BIR). In this pathway, one broken end invades the intact template and copies DNA sequences by conservative replication to the end of the template. The other broken end is lost. Except for the initial strand invasion, heteroduplex intermediates are not relevant to this pathway.

from the template and re-pairs with the other broken end. The net result of this event is a NCO in which one chromatid has a heteroduplex with a strand switch at the DSB site. The other chromatid has no heteroduplex. The same product can be formed by a double SDSA event in which the two broken ends invade different sister chromatids, synthesize DNA using these templates, and then dissociate from the sister chromatids and reanneal with each other (*Martini et al., 2011*).

In the double-strand-break-repair (DSBR) pathway, the strand displaced by DNA synthesis from the invading strand pairs with the other broken end (*Figure 1C*). Repair synthesis results in formation of a double Holliday junction (dHJ) intermediate. There are several mechanisms to resolve dHJs. First, the two junctions could be migrated toward each other, and decatenated by the Sgs1pTop3p-Rmi1p complex (dissolution). By this mechanism, there are two regions of hetDNA flanking the DSB, both located on the molecule that was originally broken.

Alternatively, the dHJ could be resolved by HJ resolvases such as Mus81p and Yen1p to form either crossover (CO) or non-crossover (NCO) products. Resolution of both HJs in the same orientation (for example, cleavage of both junctions as shown by the horizontal arrows terminated by circles) results in NCO products, whereas cleavage in different orientations (one HJ cleaved as shown with vertical arrows terminated by diamonds, and the other cleaved as shown with horizontal arrows) results in CO products (*Figure 1*). For both types of resolution, the hetDNA is on both sides of the DSB, and both the donor and recipient DNA molecule have hetDNA.

In the last pathway of HR (*Figure 1D*), break-induced replication (BIR), the invading strand results in a migrating D-loop replication structure that duplicates chromosomal sequences conservatively from the point of strand invasion to the end of the chromosome (*Donnianni and Symington, 2013*; *Saini et al., 2013*). In this pathway, the second part of the broken DNA is lost, and no heteroduplex is associated with either the donor or recombinant DNA molecule.

If the two strands of DNA that compose the heteroduplex have non-identical sequences, the heteroduplex will contain mismatches (*Figure 2*). Correction of these mismatches by the mismatch enzymes results in gene conversion, the non-reciprocal transfer of information between the two chromosomes (*Symington et al., 2014*). In *Figure 2A*, the heteroduplex has four mismatches, if these mismatches are excised from the 'blue' strand, and the resulting gap is filled in using the 'red' strand as a template, a conversion event is observed. Alternatively, if the mismatches are removed from the red strand, no conversion would be observed; such events are termed 'restorations'. A third possibility is that the mismatches would be excised from both strands resulting in 'patchy' repair. In strains that lack MMR, mismatches are not repaired and two non-identical daughter cells are produced (*Figure 2B*). In MMR-proficient strain, the patterns of heteroduplex formation may be obscured. For example, resolution of the dHJ can result in one or two chromatids that have heteroduplexes (*Figure 1C*). Following MMR, these two types of intermediates are indistinguishable because both yield the same type of conversion tracts (*Figure 2C and D*).

*Mitchel et al. (2010)* examined patterns of heteroduplex formation in recombination events between a linearized plasmid and a chromosome in an MMR-deficient strain. There were multiple sequence differences spaced about 50 bp apart within an 800 bp region of homology. By sequencing the single-colony transformants, they could detect the unrepaired mismatches that defined the length of the heteroduplex. From their analysis, they concluded: (1) most of the heteroduplexes unassociated with crossovers (NCO) were restricted to one side of the DSB site, consistent with their formation by the SDSA pathway rather than the DSBR pathway, (2) most of the heteroduplexes associated with crossovers (CO) had the patterns consistent with the DSBR pathway, and (3) most of the heteroduplexes extended less than 500 bp from the DSB site on the plasmid. In general, the observations of Mitchel *et al.* were consistent with the predictions of the model described in *Figure 1*.

In previous studies, we developed methods of detecting and mapping spontaneous and UV-induced crossovers and associated gene conversion events throughout the genome (*St Charles et al., 2012*; *St Charles and Petes, 2013*; *Yin and Petes, 2013*). Our analysis utilized diploids that were heterozygous for about 55,000 single-nucleotide polymorphisms (SNPs) and microarrays capable of distinguishing heterozygous SNPs and homozygous SNPs. One important conclusion from this analysis was that mitotic gene conversion tracts (median length of 11 kb, *St Charles and Petes, 2013*) were generally much longer than meiotic conversion tracts (median length of about 2 kb; *Mancera et al., 2008*). This observation suggests that either heteroduplexes are much longer in mitosis than in meiosis or that long mitotic gene conversion tracts involve a different intermediate such as a large double-stranded DNA gap. Our previous experiments were done in MMR-proficient strains. In the current study, using *mlh1* (MMR deficient) diploids, we show that some mitotic recombination events involve long heteroduplexes, although other events are consistent with gap repair. In addition, however, we find that recombination events initiated at a previously-described spontaneous mitotic recombination hotspot caused by an inverted pair of retrotransposons (*St Charles and Petes, 2013*; *Yim et al., 2014*) often involve long (>20 kb) double-stranded gaps rather than very

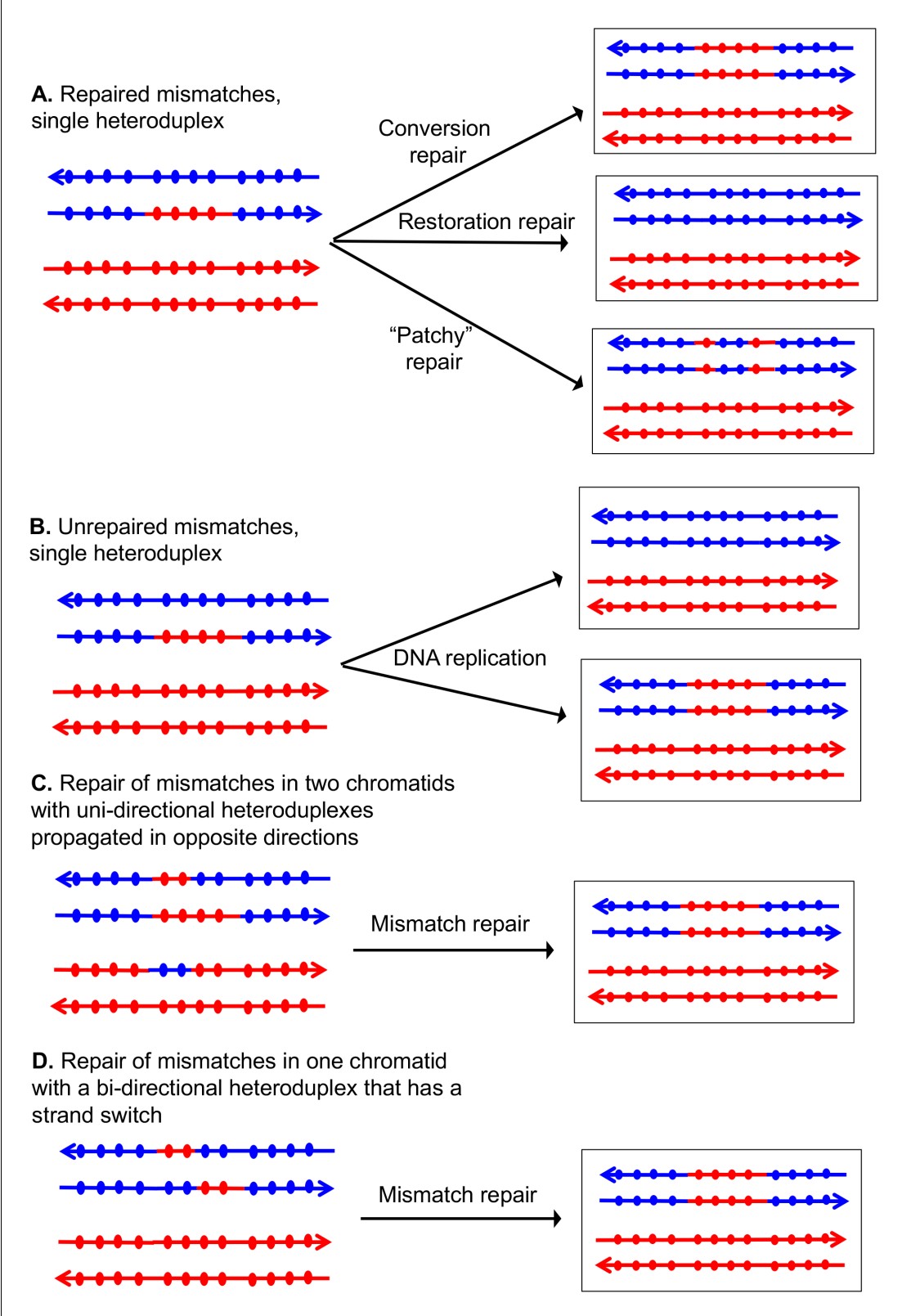

**Figure 2.** Patterns of mismatch repair in a MMR-proficient strain. Chromatids are shown as double-stranded DNA structures with circles indicating SNPs. For all parts of the figure, the blue chromatid is broken and, consequently, is the recipient of sequence information from the red chromatid. For these examples, all intermediates were resolved as NCO events. In **Figures 2A, C and D**, the products shown on the right side of each panel are a consequence of mismatch repair in the mother cell. In **Figure 2B**, the products are shown in two daughter cells following replication of the

*Figure 2 continued on next page*

*Figure 2 continued*

chromosomes of the mother cell. (**A**) Repair of mismatches within a single heteroduplex. In the top panel, the mismatches in the heteroduplex are repaired using the bottom strand as a template, resulting in a conversion event. In the middle panel, the upper strand is used as a template for MMR, resulting in a blue chromatid that is identical to an unbroken chromatid. In the bottom panel, some mismatches undergo conversion-type repair and others restoration-type repair. (**B**) Loss of mismatches as a consequence of DNA replication. Replication of a chromatid with a heteroduplex results in one product that appears to have undergone conversion-type repair, and a second that is the same as a chromatid without a recombination event. (**C**) Repair of mismatches intwo chromatids with uni-directional heteroduplexes propagated in opposite directions. If mismatches within the two heteroduplexes are repaired using the red strand as a template, a conversion event that is identical to that shown in the upper panel of *Figure 2A* would be generated. (**D**) Repair of mismatches in a one chromatid with a bi-directional heteroduplex that has a strand switch. If all mismatches are repaired using the red strand as a template, the resulting conversion product is identical to that shown in *Figure 2A and C*.

long heteroduplexes. We previously showed that most spontaneous crossovers between homologs had patterns of gene conversion that indicate that recombinogenic DSBs occur in unreplicated DNA (*Lee et al., 2009*; *Lee and Petes, 2010*). Our current study is consistent with this conclusion.

In our previous mapping of gene conversion tracts in MMR-proficient strains, we found that about 20% of the UV-induced and spontaneous events (*St Charles and Petes, 2013*; *Yin and Petes, 2013*) were 'patchy' in which markers exhibiting gene conversion flanked markers that were not converted. One simple explanation of this observation is that mismatches within one heteroduplex are sometimes repaired in a non-concerted manner by conversion-type repair and restoration-type repair. If this explanation is correct then, in the absence of repair, mismatches within a heteroduplex should be continuous (*Reyes et al., 2015*). However, in the *mlh1* strain used in this study, we find that about one-third of heteroduplexes contain mismatches flanking regions without mismatches (discontinuous heteroduplexes). Such patterns can be explained by Mlh1- independent MMR, and/or template-switching during repair synthesis. We also find recombination events that are inconsistent with the canonical DSBR models, and require additional steps such as branch migration of the Holliday junctions and/or sequential invasion of broken DNA ends.

The model of recombination shown in *Figure 1* is based primarily on studies of meiotic and mitotic recombination in yeast. Such studies often involve single meiotic recombination hotspots or mitotic recombination events induced by site-specific endonucleases. In addition, the distribution of markers (SNPs) used to analyze the events are usually sparsely distributed, and clustered near the initiating DNA lesion. Lastly, most of these studies utilized wild-type yeast strains in which some patterns of heteroduplex formation were obscured by mismatch repair. The patterns of recombination observed in our study are likely to reflect a more realistic assessment of the complex events that occur during recombination. Similar conclusions have been reached by global analyses of meiotic recombination in yeast using methods similar to those employed in our study (*Mancera et al., 2008*; *Martini et al., 2011*).

## Results

### Experimental system

The system that we previously used to analyze mitotic recombination in MMR-proficient strains is illustrated in *Figure 3*. The experimental diploids were homozygous for the *ade2–1*oc hre mutation located on chromosome XV. In the absence of a nonsense suppressor, such strains form red colonies. The chromosome arm to be assayed for recombination had a heterozygous insertion of the *SUP4-o* gene encoding an ochre suppressor located near the telomere. Diploids with zero, one, and two copies of *SUP4-o* form red, pink, and white colonies, respectively. Therefore, prior to the recombination event, cells have one copy of *SUP4-o* and form pink colonies. A cell that has a reciprocal crossover between the centromere and the *SUP4-o* marker will form a red/white sectored colony, assuming that both daughter cells have one recombinant and one non-recombinant chromosome (*Figure 3A*). In addition to the heterozygous *SUP4-o* marker, the diploids were heterozygous for about 55,000 singlenucleotide polymorphisms (SNPs) because they were constructed by mating two sequence-diverged haploids, W303-1A and YJM789 (*Lee et al., 2009*; *St Charles et al., 2012*). To determine the position of the crossover and to detect conversion events associated with the crossover, we used microarrays in which loss of heterozygosity (LOH) could be detected for about 13,000

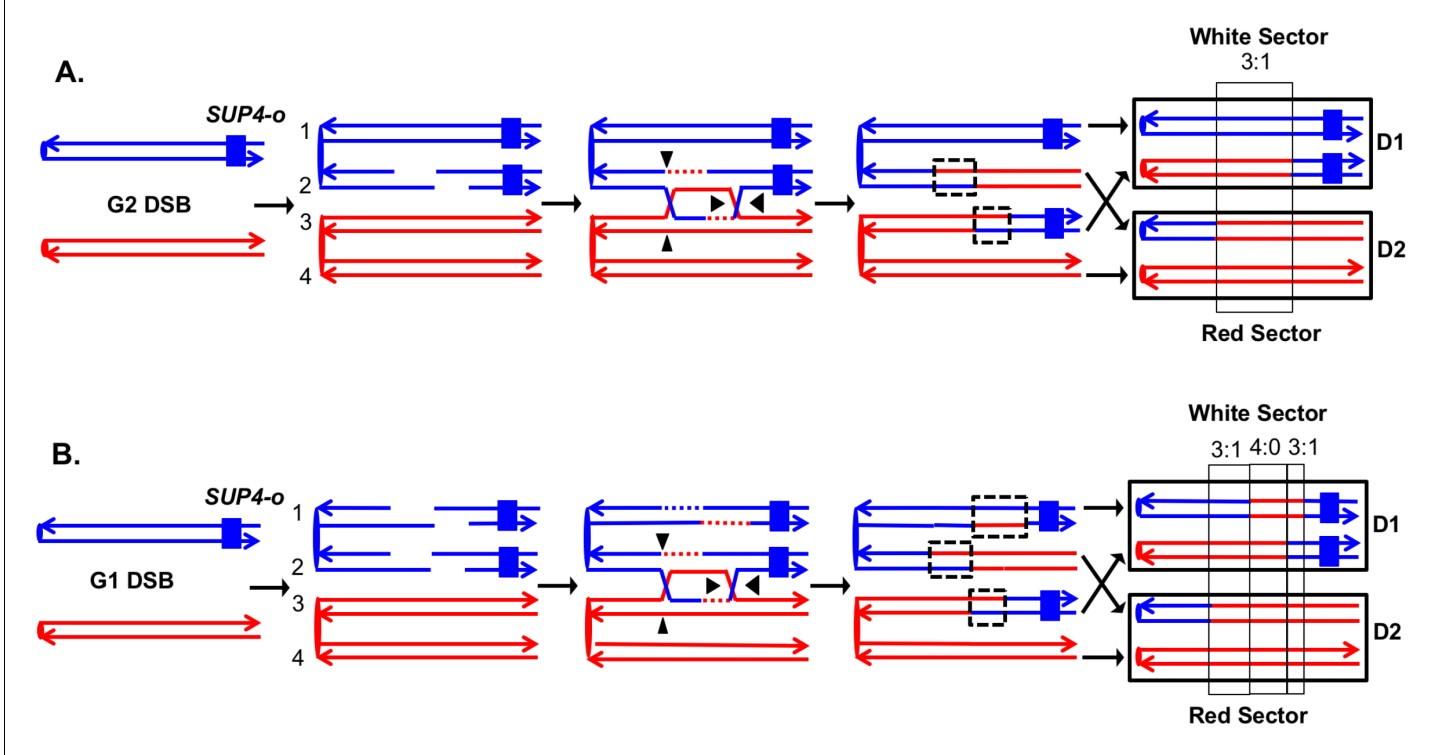

**Figure 3.** Detection and analysis of crossovers induced by DSBs in single chromatids or in unreplicated chromosomes in an MMR-proficient diploid. The lines show the two strands of each chromosome/chromatid, and ovals indicate centromeres. The strain is homozygous for *ade2-1*, an ochre allele that, in the absence of the *SUP4-o* ochre suppressor, forms a red colony. In the strains used in our study, one copy of *SUP4-o* is inserted near the telomere of one homolog. Strains with zero, one or two copies of the suppressors form red, pink and white colonies, respectively. Black triangles indicate the positions of cleavages of the dHJ. Heteroduplex intermediates are enclosed by dotted black lines, and conversion tracts are enclosed in thin black lines. Chromosomes in daughter cells are outlined by thick black rectangles, and D1 and D2 denote the two daughter cells resulting from the crossover. (**A**) Crossover initiated by a single chromatid break (SCB). Following the crossover, segregation of one recombined and one parental chromatid into each cells will generate one cell homozygous for *SUP-o* and one cell lacking *SUP4-o*; subsequent divisions will lead to a red/white sectored colony. If mismatches in both heteroduplexes undergo conversion-type repair, a 3:1 conversion event would be observed; within the boxed region, three chromatids have information derived from the red chromatid and one has information derived from the blue chromatid. (**B**) Crossover initiated by a DSB in an unreplicated blue chromosome, resulting in double sister-chromatid breaks (DSCB). DNA replication of the broken chromosome would result in two broken sister chromatids. We show the middle pair of chromatids repaired as a crossover, and the top chromatid repair by an SDSA event. In this example, we show the heteroduplex associated with the SDSA event as longer than those of the CO-associated hetDNAs. If all of mismatches are repaired as conversion events, we would see a hybrid 3:1/4:0/3:1 conversion tract. The 4:0 pattern is diagnostic of a DSCB event.

SNPs. Each SNP was represented by four 25-base oligonucleotides, two containing the Watson and Crick strands of the W303-1A allele and two containing the Watson and Crick strands of the YJM789 allele. By hybridizing genomic DNA derived from the red and white sectors, we could map the position of the crossover and associated gene conversion tract (*St Charles et al., 2012*). Details of this analysis are provided in Materials and methods.

In our previous analysis, we examined spontaneous recombination events in two types of MMR-proficient strains: those with the *SUP4-o* marker located about 100 kb from *CEN5* and those with the *SUP4-o* marker located about 1 Mb from *CEN4*. Most of the crossovers were associated with gene conversion events (solid-line boxes in *Figure 3*). Although some of the sectored colonies had the expected pattern of conversion in which one daughter cell is homozygous for SNPs located near the crossover event and the daughter is heterozygous (*Figure 3A*), more than half of the crossovers were associated with conversion events in which both daughter cells were homozygous for SNPs derived from one homolog (region marked 4:0 in *Figure 3B*). This pattern indicates that two sister chromatids were broken at the same position, consistent with a model in which the recombinogenic

DSB occurs in $G_1$, and the broken chromosome is replicated to produce two broken sister chromatids. The 4:0 tracts are often adjacent to 3:1 tracts as in *Figure 3B*. We interpret such hybrid tracts as reflecting the repair of two sister chromatids in which the extent of heteroduplex formation is different for the repair of each DSB (*Lee et al., 2009*; *St Charles et al., 2012*).

To examine mismatch-containing heteroduplexes, we constructed two diploids in the hybrid genetic background that lacked the mismatch repair protein Mlh1p (details in *Supplementary file 1* - Table S1). Both the Msh2 and Mlh1 proteins have central roles in the repair of mismatches resulting from bases misincorporated during DNA replication or formed within heteroduplexes during recombination. Elimination of these proteins elevates global mutation rates and the frequency of post-meiotic segregation to approximately the same extent (*Huang et al., 2003*; *Stone and Petes, 2006*). Both the Msh2p and Mlh1p are involved in the rejection of heteroduplexes that have closely-spaced mismatches with Msh2p having a stronger effect in most assays of this activity (*Chakraborty and Alani, 2016*). Msh2, but not Mlh1, has a role in processing branched DNA structures such as the intermediates associated with the single-strand annealing recombination pathway (*Symington et al., 2014*). We chose to examine heteroduplexes in the *mlh1* strain to avoid the possibility of losing recombination intermediates that might require Msh2p for processing. Although it is possible that some recombination intermediates could be lost in the *mlh1* strain by Msh2-dependent heteroduplex rejection, our analysis described below demonstrated that the level of mitotic recombination was not substantially affected by the *mlh1* mutation.

To detect crossovers, we used a colony-sectoring assay similar to that employed previously (*Lee et al., 2009*; *St Charles and Petes, 2013*). The heterozygous insertion of *SUP4-o* required for the sectoring assay was located near the right end of chromosome IV in diploid YYy311, and near the left end of chromosome V in diploid YYy310. In wild-type strains in which mismatches within the heteroduplex are corrected by the MMR enzymes, each sector will contain cells of only one genotype (*Figure 3*). In contrast, in the MMR-deficient strains, there may be two different genotypes in one or both sectors. In *Figure 4A*, we show the patterns of heteroduplex formation expected by the DSBR pathway in a MMR-deficient cell in which recombination initiates as a consequence of a single broken chromatid in $G_2$ of the cell cycle. Both the D1 and D2 daughter cells retain heteroduplexes with mismatches. In *Figure 4A*, the cells derived from D1 are in the white sector, and those derived from D2 are in the red sector. When the chromosomes in D1 are replicated, two different genotypes will be observed in the granddaughter cells, GD1-1 and GD 1–2. In addition, if the heteroduplexes are on different chromatids as expected for the DSBR pathway, there will also be two different genotypes in cells derived from D2.

On the left side of *Figure 4B*, we show the pattern of heteroduplexes expected for an event initiated by a DSB in an unreplicated blue chromosome (producing two sister chromatids broken at the same position as the result of DNA replication) that was repaired in $G_2$ of the cell cycle. Such cells can also produce two genotypes in each sector (*Figure 4B*). In the white sector, two types of granddaughter cells with different genotypes are shown. In the region shown in the black rectangle, GD1-1 is homozygous for the blue-derived SNPs and GD1-2 is heterozygous. In addition, in the regions defined by the green rectangles for GD1-1 and GD1-2, although both genotypes are heterozygous, the coupling of the heterozygous regions is different. In GD1-1, the blue and red SNPs within the green rectangle are on the same chromosome as the centromere-proximal blue and red SNPs, whereas in GD1-2, the blue and red SNPs within the rectangle are coupled to the centromere-proximal red and blue SNPs, respectively. As described below, for our analysis, we looked for different genotypes within sectors by methods that allowed us to examine the locations of heterozygous and homozygous SNPs, and to determine the coupling of heterozygous regions. It is important to emphasize that there is a one-to-one correspondence between the chromosomes within the sectors (labeled 1–8 on the right side of the figure) and the DNA strands in the mother cell in which the recombination event occurred (strands labeled 1–8 on the left side of the figure).

Most of our data were obtained from $G_1$-synchronized YYy310 cells treated with 15 J/m$^2$ of UV. This treatment stimulates mitotic recombination on chromosome V $> 10^3$ fold and results in about ten unselected events on other chromosomes (*Yin and Petes, 2013*). To detect cells of different genotypes within each sector of red/white sectored colonies (reflecting a crossover on chromosome V), we first purified 10–20 white and red colonies from each sector. Initially, we performed a SNP-specific microarray array of genomic DNA isolated from one of the white colonies to locate the approximate position of the recombination event. For example, we analyzed one white colony

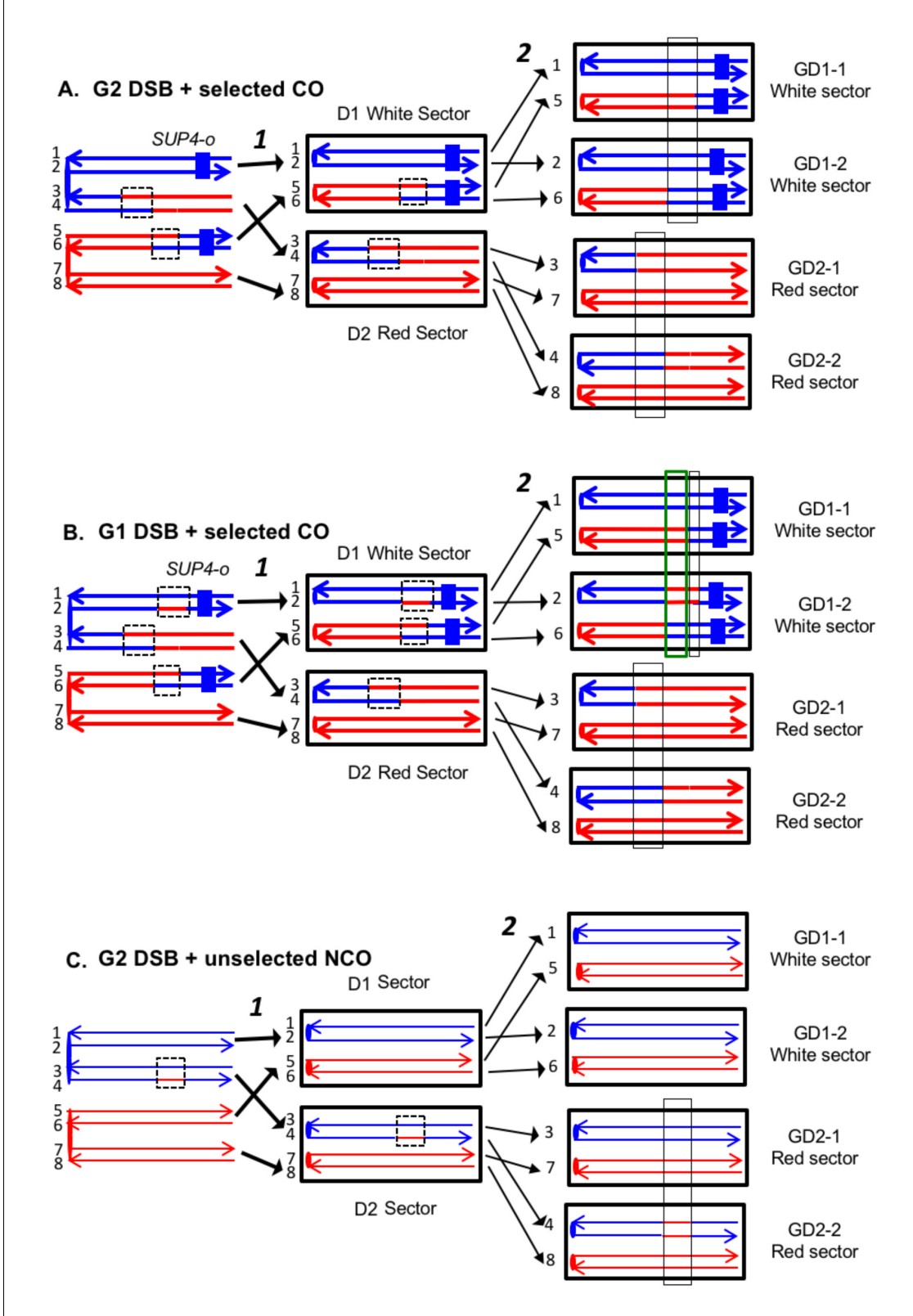

**Figure 4.** Expected recombination-associated segregation patterns of SNPs into daughter and granddaughter cells in MMR-deficient strains following a $G_2$- or $G_1$-induced DSB. We show heteroduplexes with unrepaired mismatches (outlined in dashed lines) in the daughter cells D1 and D2. Replication of unrepaired mismatches in heteroduplexes results in granddaughter cells with two different genotypes; these differences are outlined in the right side of the figure by thin black lines. The events shown in *Figure 4A and B* were selected because the crossovers between the heterozygous *SUP4-o* marker

*Figure 4 continued on next page*

*Figure 4 continued*

generated daughter cells that had two copies (resulting in a white sector) or no copies (resulting in a red sector) of *SUP4-o*. The NCO event shown in *Figure 4C* was on a homolog that did not contain the *SUP4-o* marker (shown in thin red and blue lines) and was unselected. (A) CO-associated SCB (selected crossover). If the crossover occurred by the DSBR pathway, we would expect that both sectors would have granddaughter cells with different SNP patterns. (B) CO-associated DSCB (selected crossover). As in *Figure 4A*, both sectors would have granddaughter cells with different SNPs. One distinguishing feature of the DSCB is that both granddaughter cells of one sector (GD1-1 and GD1-2) have different coupling relationships for SNPs in the same region as outlined in the green rectangle. (C) NCO-associated SCB (unselected). Strains treated with UV have many unselected events. Since these events are induced by UV at the same time as the selected event, we can use the information identifying GD1-1, GD1-2, GD2-1, and GD2-2 from the selected event to determine the patterns of heteroduplex formation for the unselected event by using whole-genome SNP arrays. In the depicted event, only one of the sectors had granddaughters with different genotypes.

(YYy310.9–5 W1) derived from a sectored red/white colony number five derived from UV-treated YYy310.9 cells. In *Figure 5*, the red and blue circles represent hybridization to W303-1A- and YJM789- derived-oligonucleotides, respectively. Hybridization values (Y-axis) are normalized such that a value of about one indicates that the experimental DNA sample is heterozygous for the W303-1A and YJM789-derived-alleles (additional details in Materials and methods); the X-axis shows Saccharomyces Genome Database (SGD) coordinates on chromosome V. LOH events that duplicate the sequences from one allele and remove sequences from the other are associated with hybridization values of about 1.5 and 0.3, respectively. Thus, by microarray analysis (*Figure 5A*), the strain (a granddaughter derived from a white sector) shown in *Figure 5A* was homozygous for YJM789-derived SNPs from the left end of chromosome V to SGD coordinate 51915. The strain was heterozygous for SNPs between 53612 to 54198, and then homozygous for W303-1A-derived SNPs from coordinates 56117 to 57170. Lastly, the strain was heterozygous for SNPs between coordinates 60701 and *CEN5*. As will be discussed below, this strain represents one of the two granddaughter genotypes observed in the white sector.

Based on this information, we examined genomic DNA from eight white colonies and eight red colonies by PCR analysis to look for different genotypes within one sector. We chose to examine SNPs that were near the transitions of heterozygous and LOH regions. For example, at position 51707, there is a polymorphism that distinguishes the YJM789- and W303-1A-derived SNPs (*Lee et al., 2009*). In the W303-1A genome, this polymorphism is part of a restriction enzyme recognition site for *Dra*I that is absent in YJM789. Consequently, we PCR-amplified the region containing this site from the individual white and red colonies, treated the resulting fragments with *Dra*I, and examined the resulting products by gel electrophoresis. Five of the white colonies were homozygous for the YJM789-form of the SNP, and three were heterozygous, demonstrating that this SNP was included as an unrepaired mismatch in one of the daughter cells. By a similar approach, we showed that SNPs heterozygous in the starting strain at coordinates 54915, 56166, and 57448 were homozygous for the W303-1A-form of the SNP in three red colonies, and heterozygous in the remaining five. Thus, this analysis allowed us to unambiguously define two different granddaughter genotypes in both the white (GD1-1, name shortened to W1 in *Supplementary file 2*; GD1-2, name shortened to W2) and red (GD2-1, name shortened to R1; GD2-2, name shortened to R2).

The complete analysis of the granddaughter genotypes required two additional steps. First, we mapped one representative of each granddaughter genotype using whole-genome SNP arrays. This analysis allows us to map heterozygous and homozygous SNPs on chromosome V and unselected events throughout the genome. The coordinates for transitions between heterozygous and homozygous SNPs for the UV-treated YYy310 samples are in *Supplementary file 1* - Table S2. An example of the SNP microarray analysis for one granddaughter diploid strain derived from the white sector (YYy310.9–5 W1) is shown in *Figure 5A*.

Many of the granddaughter isolates, such as that shown in *Figure 5A*, had two or more heterozygous regions. To examine the coupling relationships of the heterozygous regions, we sporulated the four granddaughter diploids derived from each sectored colony, and dissected tetrads. One spore derived from each strain was then examined by microarrays. The pattern shown in *Figure 5B* represents the analysis of one haploid spore derived from granddaughter W1. This pattern shows the arrangement of SNPs on the top chromosome of *Figure 5C*, and the arrangement of SNPs on the other homolog (bottom chromosome of *Figure 5C*) can be directly inferred.

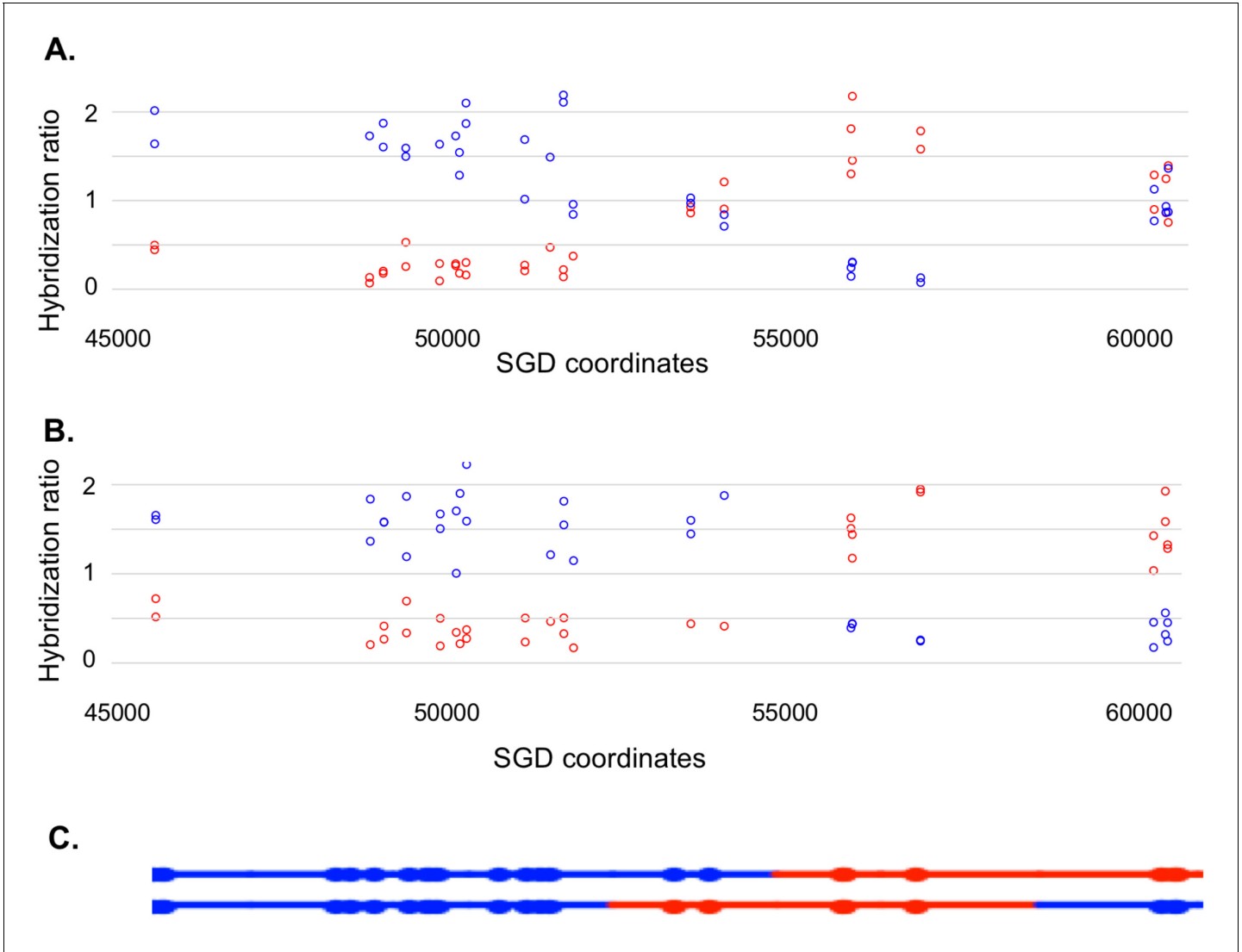

**Figure 5.** Microarray analysis of one granddaughter cell in a sectored colony resulting from a recombination event on chromosome V. By methods described in the text, we identified two granddaughter genotypes associated with the red and white sectors of the YYy310-9-5WR sectored colony. SNP-specific microarrays were done on genomic DNA isolated from all four granddaughter strains, and from spore derivatives of each of these granddaughters. (A) Microarray analysis of the W1 granddaughter strain. Genomic DNA was isolated, and hybridized to SNP-specific microarrays in competition with differentially-labeled genomic DNA from a heterozygous control strain (details in Materials and methods). The Y axis shows the hybridization ratio of the experimental samples to allele-specific SNPs normalized to the hybridization levels of the control heterozygous sample; blue and red circles indicate hybridization ratios to YJM789- and W303-1A-specific SNPs, respectively. Heterozygous samples have a ratio of about 1, samples in which the strain-specific allele is present in two copies have a ratio of about 1.7, and those in which the strain-specific allele is missing have a ratio of about 0.3. The X axis has SGD coordinates for chromosome V in the region of the recombination event. Between the left telomere (coordinate 1) and coordinate 51915, the W1 granddaughter is homozygous for the YJM789-derived SNPs, and heterozygous for SNPs between coordinates 60701 and the right telomere. In summary, chromosome V in this strain has a large terminal region that is homozygous for YJM789 SNPs, a short region of heterozygous SNPs, a short region that is homozygous for W303-1A SNPs, and a large heterozygous region that makes up the remainder of the chromosome. (B) Microarray analysis of a spore derived from the W1 granddaughter strain. The W1 diploid was sporulated, and dissected. We examined the genomic DNA of one spore by SNP microarrays. From the pattern of SNPs in the spore, we conclude that one homolog in W1 has blue SNPs for the heterozygous region located near coordinates 55000 and red SNPs for the heterozygous region located between coordinates 60000, and the right telomere. The other homolog has the reciprocal pattern. (C) Inferred arrangement of red and blue SNPs on the two homologs of W1 based on the microarray results shown in **Figure 5A and B**. This pattern is reproduced in the top part of **Supplementary file 2** - S8.

This analysis allowed a complete determination of the arrangement of SNPs in W1, W2, R1, and R2, and was performed for all sectored colonies. An example of how the patterns of LOH in the granddaughter cells can be used to figure out the pattern of heteroduplexes in the mother cell is shown in *Figure 6*. Each chromosome in the granddaughter cells has the same pattern of SNPs as one of the DNA strands in the daughter cell. The source of the centromeres determines which of the chromosomes in the granddaughter cells was derived from which chromosome in the daughter cell. For example, in *Figure 6*, the chromosomes labeled 1 and 2 have red centromeres, and were derived from the daughter chromosome with the red centromere. Chromosomes 5 and 6 were derived from the daughter chromosome with the blue centromere. Similarly, both the daughter chromosomes labeled 1 and 2 from the white sector and the daughter chromosomes labeled 3 and 4 from the red sector have red centromeres, and were connected as a pair of sister chromatids to the red centromere in the original mother cell.

The transitions between regions of homozygous and heterozygous SNPs for all events are given in *Supplementary file 1* - Table S2, and *Supplementary file 1* - Table S3 summarizes the classes of these events as crossovers (CO) and non-crossovers (NCO). An important feature of our analysis is that once we have defined daughter and granddaughter cells based on the crossover on the selected chromosome, this lineage information also can be applied to unselected CO and NCO events on other chromosomes. For example, although the NCO event shown in *Figure 4C* does not generate a sector, our whole-genome microarray analysis of granddaughter cells defined by the selected crossover allows us to detect and to fully describe the unselected event.

Our analysis of hetDNA included three types of experiments. First, we examined seven sectored colonies derived from the UV-treated YYy310 strain in which the *SUP4-o* marker was inserted near the left end of chromosome V. In addition to the selected crossover on chromosome V, these strains had about ten unselected events (both crossovers and conversions unassociated with crossovers) per strain because of the recombinogenic effects of UV (*Yin and Petes, 2013*). Second, we analyzed five spontaneous crossovers on chromosome V in YYy310. Lastly, we characterized six spontaneous crossovers on chromosome IV in strain YYy311 (heterozygous for the *SUP4-o* marker on the right end of chromosome IV) that were associated with the HS4 recombination hotspot.

## Analysis of selected crossovers on chromosome V and unselected recombination events induced by UV in YYy310

We found previously (*Yin and Petes, 2013*) that more than half of the recombination events detected in $G_1$-synchronized diploids treated with high doses of UV had the pattern of recombination indicative of the repair of a $G_1$-associated DSB (*Figure 3B*). In addition, DSBs induced by UV were detected by gel electrophoresis (*Covo et al., 2012*). Such DSBs could be formed as a consequence of nucleotide-excision-repair (NER) enzymes acting on two very closely-spaced lesions on opposite strands or as the result of expansion of short single-stranded NER-generated gaps on opposite strands by the action of Exo1p. Cells treated with low doses of UV (1 J/m$^2$) primarily had patterns of gene conversion (3:1 events) indicative of a single broken chromatid (*Figure 3A*). Based on our analysis of UV-induced LOH events in wild-type and *rad14* (NER-defective) cells (*Yin and Petes, 2015*), single-broken chromatids are likely generated two different ways: by replication of a chromosome with an NER-generated gap, and by Mus81-dependent processing of DNA structures formed when a replication fork is blocked by an unprocessed UV-generated lesion.

We examined seven sectored colonies derived from two isogenic independently-constructed YYy310 diploids (YYy310-9 and YYy310-10). By the methods described above, we identified two different genotypes within each sector in all seven sectored colonies; these genotypes are designated W1 and W2, and R1 and R2 for the white and red sectors, respectively. Our diagnosis of granddaughter genotypes was initially based on PCR analysis of SNPs associated with the selected crossover on chromosome V. For five of the seven sectored colonies, this analysis was sufficient to detect W1, W2, R1, and R2. For the two sectored colonies in which we could not identify two genotypes in the sectors using markers on V, we used unselected events on other chromosomes to identify the granddaughters.

In addition to the seven selected crossovers, there were 71 unselected crossover or gene conversion events among the sectored colonies. Interpretations of eight classes of relatively simple recombination events (described in detail below) are shown in *Supplementary file 2* - S1 and S2. In *Supplementary file 2* - S3-S80, all of the selected and unselected events are shown, with the upper

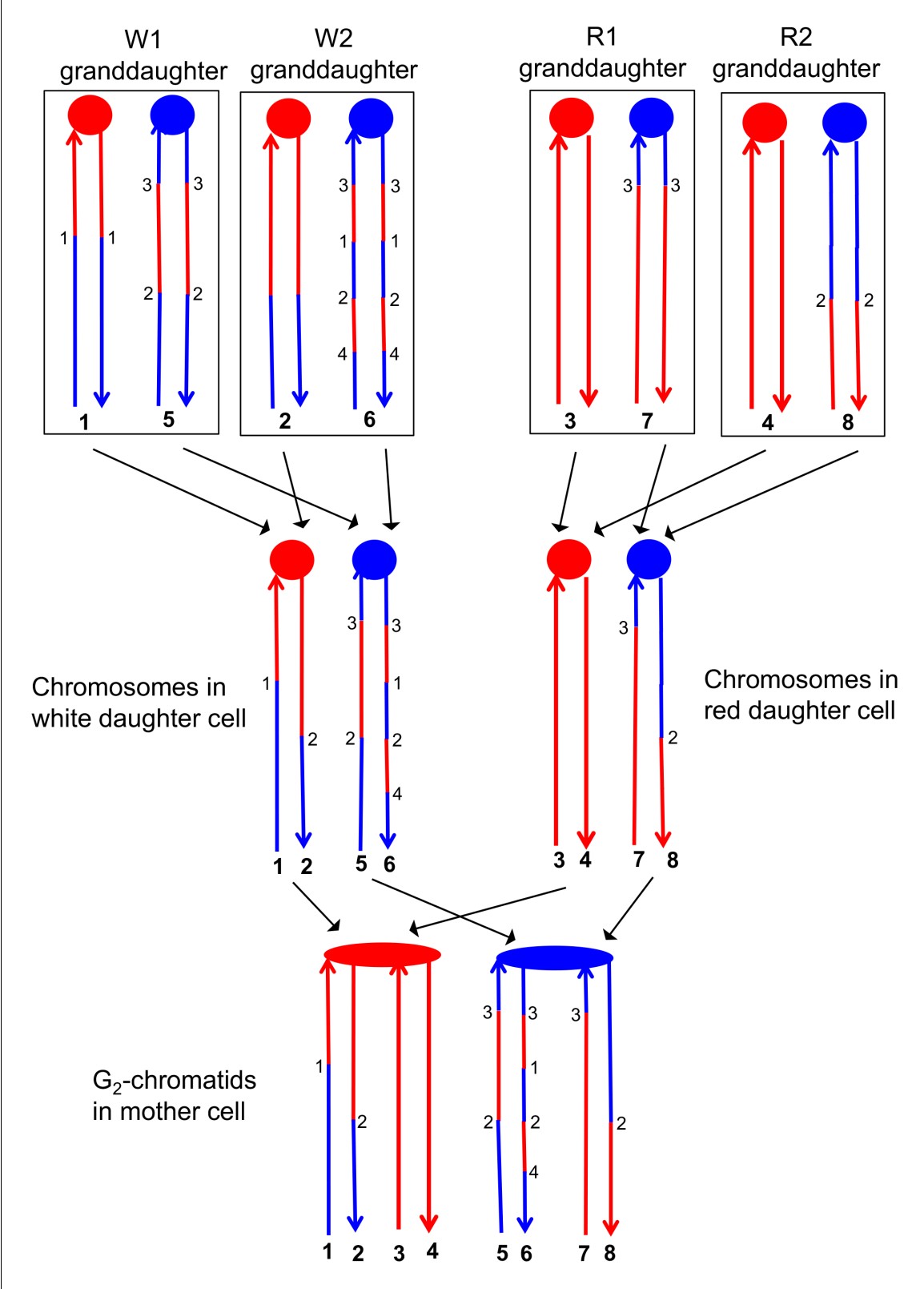

**Figure 6.** Correlation of SNP patterns in granddaughter cells with heteroduplex patterns in the mother cell. The example shown is the same crossover on chromosome V described in *Figure 5* (YYy310-9-5WR sectored colony). Each chromatid/chromosome is shown as a double-stranded DNA molecule with ovals and circles showing the centromeres. As in *Figure 5*, red and blue lines signify sequences derived from W303-1A and YJM789, respectively. By the methods described in the text, we identified two genotypes within each sector. The microarray analysis illustrated in *Figure 5* allowed us to

*Figure 6 continued on next page*

*Figure 6 continued*

determine the patterns of LOH in each granddaughter cell (top of the figure). Each chromosome in the granddaughter cell was derived by replication by chromosomes in the daughter cells (middle of the figure). Chromosomes with the same color of centromere in the granddaughter cells must derive from a daughter chromosome with a centromere of the same color. Numbers in bold represent chromatids (top part of figure) or individual DNA strands (bottom part of the figure). For example, chromatids 1 and 2 in the W1 and W2 granddaughters must represent the strands labeled 1 and 2 in the white daughter cell. A similar procedure can be used to assign the strands to the chromosomes in all of the daughter cells. The chromosomes in the daughter cells represent segregants of the $G_2$ chromatids in the mother cell. The daughter chromosomes with red centromeres must have been derived from the paired sister chromatids with the red oval centromere, and a similar conclusion can be drawn about the blue centromeres of the daughter chromosomes and the oval blue centromere in the mother cell. By the pattern of SNPs, we can also conclude that the chromatids in the mother cell with strands 1 and 2, and 7 and 8 were involved in a CO, whereas the other two chromatids were involved in the NCO mode of repair. Numbers that are not in bold represent transitions between W303-1A-derived SNPs and YJM789-derived SNPs; transitions with the same number have identical breakpoints with the resolution of our microarray analysis.

part of the figure depicting the pattern of heterozygous and homozygous SNPs on the two chromosomes of the granddaughter of the white (W1 and W2) and red (R1 and R2) sectors. The bottom part of each figure shows the inferred patterns of heterozygous and homozygous SNPs in the mother cells, prior to the segregation of chromosomes into the daughters. Each chromatid is depicted as double stranded DNA and circles of different colors at the same positions on the two strands indicate unrepaired mismatches in heteroduplexes. In the discussion below, SNPs from the W303-1A-derived homolog will be called 'red SNPs' and those from the YJM789-derived homolog will be called 'blue SNPs', consistent with the figures. Below, we summarize some of our findings based on this complex dataset.

## SCB and DSCB events are approximately equally frequent

First, we can divide the events into two major classes, those in which the event was initiated by a single broken chromatid in S or $G_2$ (SCB, single chromatid break) and those initiated by two broken chromatids (DSCB, double-sister-chromatid break), likely reflecting replication of a chromosome broken in $G_1$. The heteroduplex patterns inferred from our analysis of granddaughter cells for two events are shown in *Figure 7*. In the event shown in *Figure 7A*, only chromatid two has a heteroduplex, suggesting that the initiating event was a SCB. In contrast, in *Figure 7B*, chromatids 1 and 3 have the CO configuration of markers, whereas chromatid four has a NCO configuration. This observation suggests that this event was initiated by a DSCB. Of 77 events that could be unambiguously classified (*Supplementary file 1* - Table S3), there were 39 SCB and 38 DSCB events. In previous studies of events in wild-type strains treated with the same dose of UV (*Yin and Petes, 2013*), we also found approximately equal frequencies of these two classes (*Supplementary file 1* - Table S6).

Our conclusion that many of the observed recombination events, both spontaneous and UV-induced, reflect a $G_1$-initiated DSB is surprising, but consistent with our previous studies (*Lee et al., 2009*; *St Charles and Petes, 2013*; *Yin and Petes, 2013*). In considering this conclusion, a number of points need to be emphasized. First, we propose that DSCB events result from replication of a broken chromosome. This DSB cannot be efficiently repaired in $G_1$ because the enzymes involved in end resection and resolution of recombination intermediates are not active in $G_1$ (*Symington et al., 2014*). An alternative pathway to DSB repair in $G_1$ cells is non-homologous end-joining (NHEJ). In yeast, however, this pathway is inefficient (*Siede et al., 1996*). In addition, when we previously compared the frequency of mitotic crossovers in diploids that were heterozygous at the mating type locus (a condition that represses NHEJ, *Kegel et al., 2001*) or hemizygous (active NHEJ), there was no consistent difference in the frequency of mitotic crossovers and the ratios of $G_1/G_2$ events in the two types of strains (*Barbera and Petes, 2006*; *Lee et al., 2009*; *Yin and Petes, 2013*).

Since the repair of the broken chromosome in $G_1$ is inefficient, we suggest that the broken chromosome is replicated to produce two chromatids that are broken at the same positions. Since the equivalent location of the breaks precludes repair by sister-chromatid recombination, they are repaired by interaction with the chromatids of the intact homolog, resulting in the observed LOH events. It is important to emphasize that DSCBs may actually represent a less common DNA lesion than SCBs. Our system, however, detects only those events that lead to LOH. Repair of DSBs by recombination between sister chromatids is undetectable by our system. Using a different genetic

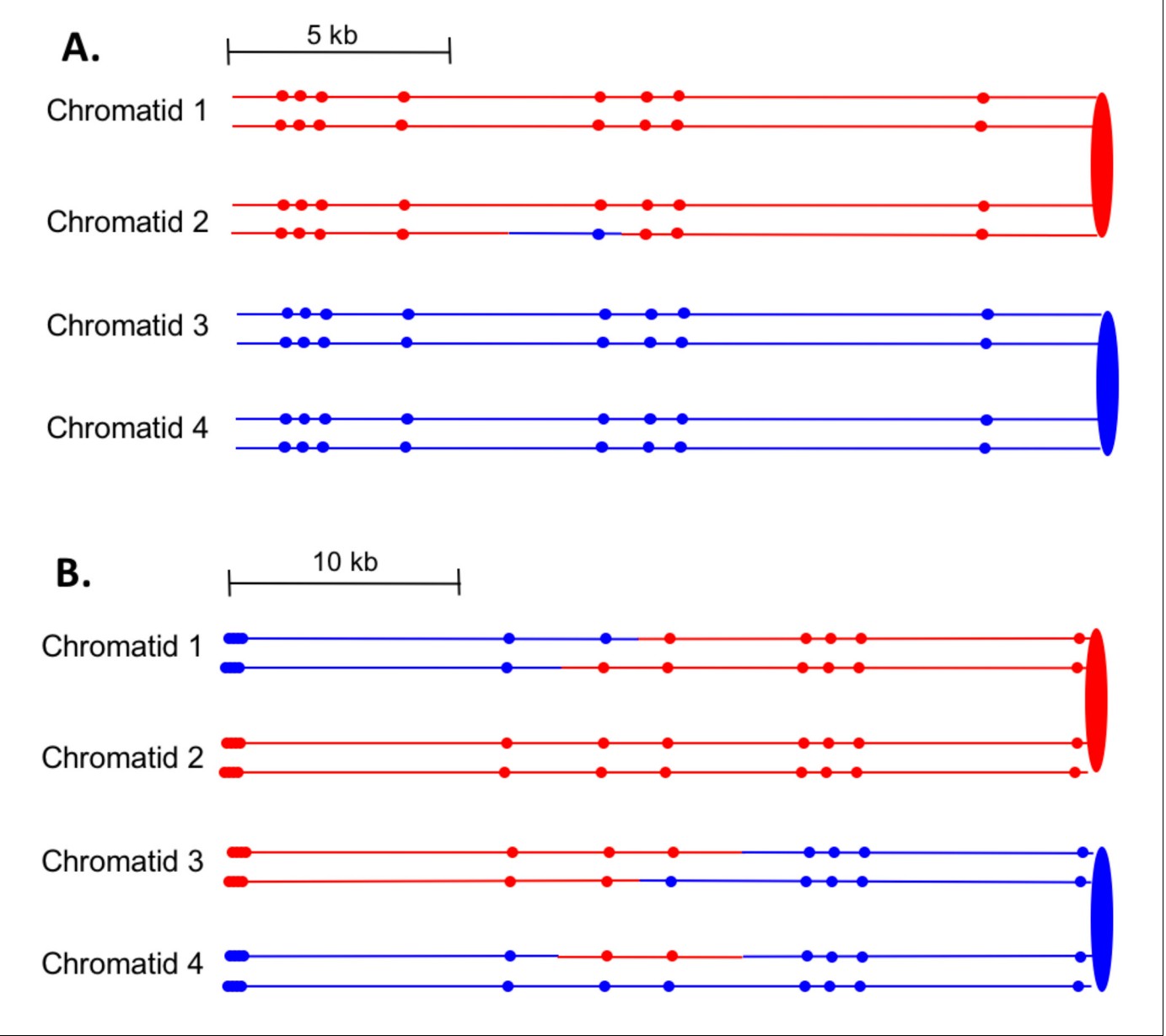

**Figure 7.** Examples of heteroduplex patterns associated with SCB- and DSCB-initiated events. Both the SCB event shown in 7A and the DSCB event shown in 7B were unselected events from the same sectored colony (YYy310-905WR; *Supplementary file 1* - Table S3). The 7A event occurred on chromosome IV (labeled IV-2 in *Supplementary file 1* - Table S3), and the 7B event was on chromosome XIV. The distances between SNPs (shown as red or blue circles) are to the scale shown above each set of chromatids. (**A**) Inferred pattern of heteroduplexes in an SCB event. The patterns of markers in the granddaughter cells that define the patterns inferred in the $G_2$ mother cell are in *Supplementary file 2* - S6. (**A**) Inferred pattern of heteroduplexes in an DSCB event. The patterns of markers in the granddaughter cells that define the patterns inferred in the $G_2$ mother cell are in *Supplementary file 2* - S16.

system, *Kadyk and Hartwell (1992)* concluded that > 90% of the DSBs generated in $G_2$-synchronized yeast were a consequence of sister-chromatid recombination rather than inter-homolog exchange; we observed a similar ratio for spontaneous events (*Zhao et al., 2017*). Although the current analysis is restricted to examining those events that lead to LOH, LOH events are critical for the development of some classes of tumors (*Cavenee et al., 1985*).

In addition to the DSCB events, we observed a substantial number of SCBs in UV-treated cells. Such events likely have two sources. Although most small single-stranded gaps resulting from NER

are likely repaired in G$_1$, some may persist into S. Replication of a nicked template would result in an SCB. An alternative source of SCBs may be unexcised dimers that stall the replication fork, resulting in an S-phase-associated DSB (*Yin and Petes, 2015*).

Excluding the selected CO events on chromosome V, of the 33 DSCB events, 22 had two NCOs, nine had one NCO and one CO, and two had two COs. For our mechanistic interpretation of these events (to be described in detail below), we consider the two repair events resulting from the DSCB separately. The chromatids involved in a CO are almost always clear since the terminal SNPs at each end of the chromatid are altered in their coupling relative to the centromere as described above for chromatids 1 and 3 of *Figure 7B*. For NCO events that have a single recombinant chromatid, we cannot determine which of the two non-recombinant chromatids was the donor of sequence information. For example, in *Figure 7A*, the donor chromatid could be either chromatid 3 or four since these chromatids are identical. For such events, in *Supplementary file 1* - Table S3, we arbitrarily assigned one chromatid as the donor.

It should be emphasized that in our experiments, as in prior studies (*Lee et al., 2009*; *St Charles and Petes, 2013*; *Yin and Petes, 2013*), the DSCB events are far too frequent to represent two independent SCB events. The frequency of red/white sectored colonies (reflecting a crossover on chromosome V) in the UV-treated YYy310 strain was about 1.5%. About half of these events are a consequence of SCB. The likelihood of two independent SCB events resulting in an apparent DSCB is about (0.75%)$^2$ or $5.6 \times 10^{-4}$; we observed two orders of magnitude more DSCB events than this calculated frequency. In addition, the calculated frequency was based on double events occurring anywhere within the 120 kb between the *SUP4- o* insertion and *CEN5*. Most of the DSCB events involve DSBs that occur within a few kb of each other. Lastly, half of the time, two independent events would have donor chromatids derived from different homologs; only three of the DSCB events involved different homologs. Lastly, we point out that our measurement of the frequency of observed SCBs is an underestimate of the formation of SCBs, since many of the SCBs are likely to be repaired by sister-chromatid recombination (*Kadyk and Hartwell, 1992*).

Many previous studies of recombination in yeast show that the chromosome with the initiating DNA lesion acts a recipient of sequence information transferred from the intact donor chromosome (*Symington et al., 2014*). From our analysis, we can determine whether the homologs derived from the haploid parents YJM789 and W303-1A were equally susceptible to the initiating DNA lesion. As shown in *Supplementary file 1* - Table S3, of the 38 DSCB events, 18 were initiated on the YJM789-derived homolog and 20 were initiated on the W303-1A-derived homolog. Of 40 SCBs (the event shown in *Supplementary file 2* - S60 had two SCBs), the numbers initiated on the YJM789 and W303-1A homologs were 15 and 25, respectively; by chi-square analysis, these numbers are not significantly different from a 1:1 ratio (p=0.16). As expected, therefore, both homologs are equally susceptible to recombinogenic UV-induced DNA lesions.

## Location of the DNA lesion that initiates recombination

For some events, the pattern of SNPs in recombinant chromatids allowed us to predict locations of the initiating DNA lesion, although most of these predictions had a degree of uncertainty. For single SCB events, the predicted position on the chromatid with the recombinogenic DNA lesion (likely a DSB) is shown as an arrow labeled DSB1 in *Supplementary file 2* - S3-S80. For example, in *Supplementary file 2* - S4, we show an arrow to the left of the conversion tract in chromatid 3. With equal validity, the arrow could have been placed to the right side or in the middle of the conversion tract. For DSCB events that are explicable by a single DSB on an unreplicated chromosome (*Supplementary file 2* - S15), we show the DNA lesions by labeled arrows at the same positions on the replicated chromatids. In events that appear to require two independent DNA lesions (for example, *Supplementary file 2* - S3), DSBs are labeled DSB1 and DSB2. For some DSCB events, we used information for all the recombinant chromatids to infer the position of the initiating DSB. For example, in *Supplementary file 2* - S9, the initiating lesion for the NCO event on chromatid three could be placed at either end or in the middle of the conversion/heteroduplex tract. The pattern of the heteroduplex tract in chromatid 4, however, suggested the location of the initiating DSB shown by the arrow in *Supplementary file 2* - S9.

## Simple classes of CO and NCO chromatids

The 116 UV-induced CO and NCO events in YYy310 were classified into nine groups (*Supplementary file 1* - Table S3), Classes 1–8 representing events with relatively straightforward interpretations (71 events) and a 'Complex' class requiring more complicated mechanisms (45 events). In our discussion of mechanisms, we will assume that the recombinogenic lesion is a DSB, although there is evidence that DNA nicks are also recombinogenic (*Fabre et al., 2002*; *Lettier et al., 2006*; *Davis and Maizels, 2014*). As described above, the DSCB events are most easily explained as reflecting a DSB in $G_1$; the SCB events could be initiated by either a DSB in $S/G_2$ or a DNA molecule that is nicked in $G_1$ and replicated to generate the DSB in $S/G_2$.

In the discussion of classes below, we use the term 'heteroduplex' to refer to duplex regions with mismatches, indicating DNA strands derived from different homologs. Conversion tracts are regions in which both of the interacting chromatids have homoduplexes derived from the donor chromatid (*Figure 2*), and restoration tracts are regions in which the mismatches in heteroduplexes were inferred to be corrected to restore the original sequence of the recipient chromatid. The eight classes of events (*Figure 8*, Classes 1–4 in *Supplementary file 2* - S1 and Classes 5–8 in *Supplementary file 2* - S2) are: Class 1 (NCO, single continuous heteroduplex tract on one side of putative DSB site), Class 2 (NCO, single continuous conversion tract on one side of putative DSB site), Class 3 (NCO, hybrid heteroduplex/conversion tract located on one side of putative DSBs site), Class 4 (CO, heteroduplex on only one side of putative DSB site), Class 5 (NCO, heteroduplex on one side of putative DSB site, and conversion tract on the other), Class 6 (CO, uni-directional heteroduplexes on both chromatids propagated in opposite directions), Class 7 (CO, no observable heteroduplex or conversion tract on either recombinant chromatid), Class 8 (NCO; two recombinant chromosomes, one containing a conversion tract, and the second with a heteroduplex involving SNPs at the same position). The number of events in each of these classes are shown in *Table 1*.

Before discussing these classes in detail, we point out that heteroduplexes can be detected only if the heteroduplex includes a SNP. The diploid used in our study is heterozygous for about 55,000 SNPs, but only 13,000 of these SNPs are included on the microarrays (*St Charles et al., 2012*). Since the average distance between SNPs, as assayed by the microarray, is about 1 kb, a fraction of short heteroduplex/conversion tracts will be undetectable (*St Charles et al., 2012*; *Zheng et al., 2016*). We estimated the fraction of undetected events using the same procedure as *Mancera et al. (2008)* and *Martini et al. (2011)*. Assuming that all crossovers are associated with heteroduplexes on both interacting chromatids, we determined what fraction of such chromatids had an observable heteroduplex or conversion (Table S3). Of 33 crossovers, 23 had regions of heteroduplex on both recombined chromatids, eight had only one region of heteroduplex on the two chromatids, and two had no detectable heteroduplex on either chromatid. Thus, we estimate that our methods detect about 80% (54/66) of the conversion events that are unassociated with crossovers. This calculation is dependent on the assumption that the lengths of the crossover-associated conversions are similar to those of the conversions that are unassociated with crossovers.

The largest class, Class 1, has the pattern of SNPs expected for events initiated by the SDSA pathway (*Figure 1A*, *Figure 8A*). Class two events could also reflect SDSA events, however, the observed conversion tracts are not consistent with the simplest form of the DSBR model. One possibility is that, following disengagement of the invading end, the mismatches in the heteroduplex are repaired by an Mlh1-independent pathway (*Figure 8B*). *Coïc et al. (2000)* reported the existence of a short-patch (<12 bp) Msh2-independent repair pathway in yeast. This pathway was hypothesized to explain a small number of recombination events in *msh2* strains in which regions of unrepaired mismatches were interspersed with repaired mismatches. No genes in this pathway have been identified, although *Coïc et al. (2000)* argued that the nucleotide excision repair proteins were not involved. Alternatively, short-patch mismatches close to the DSB could be removed by the proofreading activity of polymerase δ (*Anand et al., 2017*). Although such pathways could explain conversion tracts limited to one or a few closely-spaced SNPs (*Supplementary file 2* - S4, for example), it seems unlikely to explain conversion tracts extending over several kb (*Supplementary file 2* - S23: for example).

An alternative model is that the Class 2 NCO events are a consequence of the repair of a double-stranded DNA gap (*Figure 8C*). In experiments in which gapped plasmids are transformed into yeast, gap-repair occurs readily (*Orr-Weaver et al., 1981*), and such gap repair was an intrinsic part

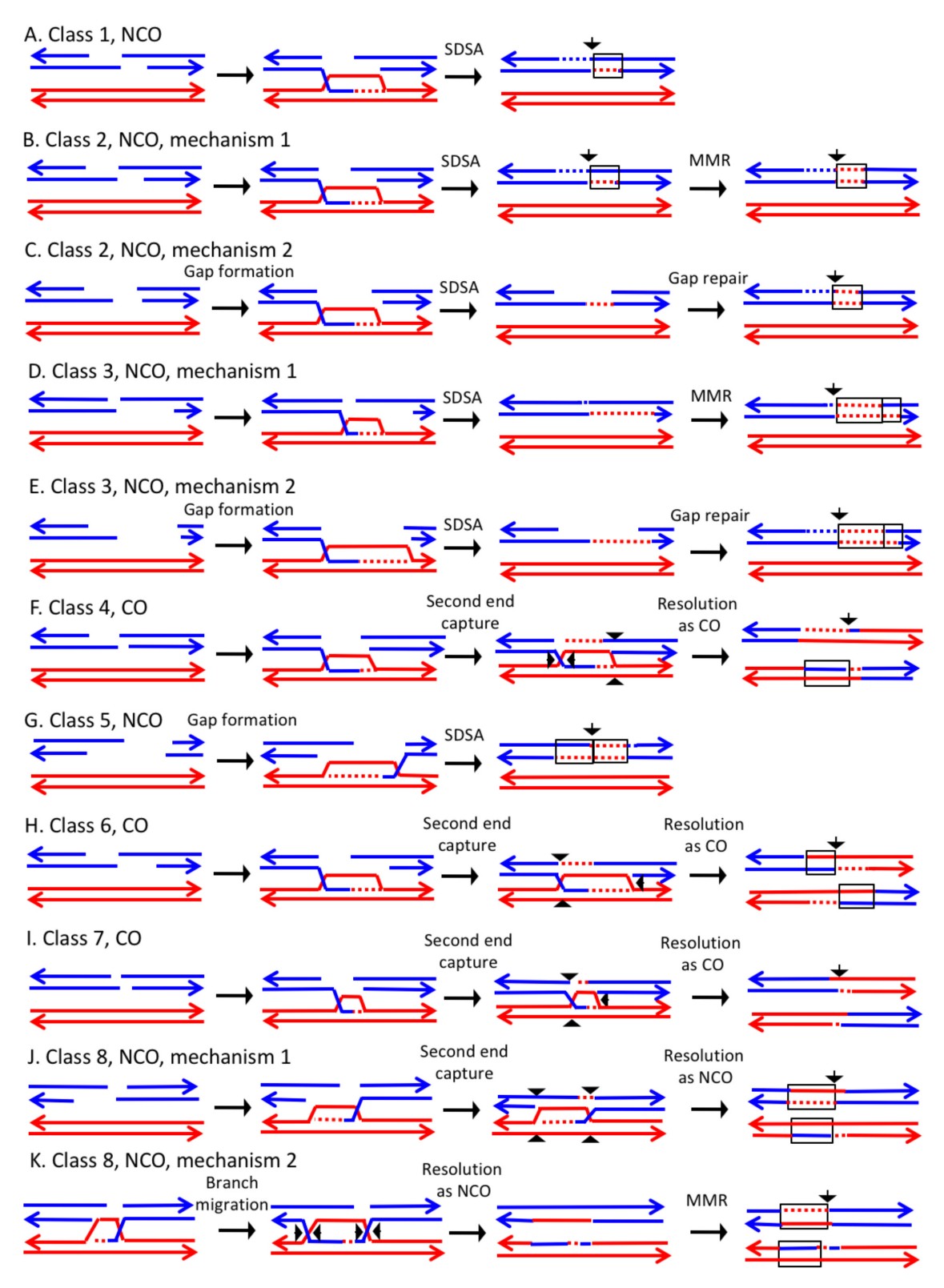

**Figure 8.** Mechanisms that generate Classes 1–8 recombination events. For all events, we show the initiating DSB on the blue chromatid. Dotted lines show sequences generated by replication or during mismatch repair. Black arrows indicate the position of the initiating DSB. The mechanisms are discussed in detail in the main text. Red and blue lines represent DNA strands of the W303-1A-derived chromatid and YJM789-derived chromatid, respectively. Arrows on these strands indicate the 3' ends. (**A**) Class 1. Class one events are NCOs formed by one-ended SDSA. (**B**) Class 2. These NCO

*Figure 8 continued on next page*

*Figure 8 continued*

events could be generated by one-ended SDSA, followed by Mlh1p-independent MMR. (**C**) Class 2. An alterative possibility is that Class two events reflect repair of a double-stranded DNA gap. (**D**) Class 3. In this NCO class, the heteroduplex region is adjacent to a conversion tract. Such events could reflect a heteroduplex tract in which mismatches are repaired in one part of the tract and left unrepaired in the other. (**E**) Class 3. An alternative model for this class is that the conversion tract is the result of repair of a double-stranded DNA gap with a heteroduplex region at one end. (**F**) Class 4. For this CO class, a heteroduplex is observed on one chromatid but not the other. This class could be explained by the DSBR pathway in which heteroduplex region is short relative to the other; if the short heteroduplex does not contain a mismatch, it would be undetectable. (**G**) Class 5. In this NCO class, the heteroduplex region is located on the opposite of the DSB site from the conversion region. This pattern is consistent with the repair of a double-stranded DNA gap that was restricted to one of the broken ends. (**H**) Class 6. This CO class is identical to the pattern expected for the DSBR model. (**I**) Class 7. In this CO class, no heteroduplexes or conversion tracts are observed adjacent to the crossover, consistent with the formation of a dHJ with short heteroduplex tracts that do not include mismatches. (**J**) Class 8. In this NCO class, one chromatid has a conversion tract, and the other chromatid has a heteroduplex involving SNPs at the same position. This event could reflect resolution of a dHJ intermediate in the NCO model in which one region of heteroduplex is undetectable. (**K**) Class 8. An alternative mechanism involves branch migration of a HJ, followed by resolution of the intermediate in a NCO mode. Mismatches in one of the two chromatids are repaired to generate a conversion tract.

of the original DSBR model (*Szostak et al., 1983*). Since in most meiotic recombination events in yeast, DSBs are processed by degradation of only one strand of the duplex (*de Massy et al., 1995*), gap repair is usually not considered part of the standard DSBR model. *Giannattasio et al. (2010)*, however, reported that the Exo1p could cause expansions of nucleotide excision repair tracts from 30 bases to several kb. Based on our previous observations that showed a two-fold reduction in UV-induced mitotic recombination in the *exo1* mutants, we previously suggested that Exo1-mediated tract expansion has a role in generating recombinogenic lesions in UV-treated cells (*Yin and Petes, 2014*); the Exo1p could be involved in the long-tract Class two events. In summary, we suggest that events with long conversion tracts may reflect the repair of a double-stranded DNA gap, and short regions of conversion may represent Mlh1-independent MMR. Another possible mechanism for Class two events is that the broken end invades the homolog, and copies sequences by a BIR-like mechanism before disengaging from the homolog and re-engaging with the other broken end.

Class 3 NCO events are similar to Class 2, and have a similar mechanistic explanation. For Class three events, however, mismatches in one segment of the heteroduplex are repaired and those in the adjacent segment are not (*Figure 8D*). Alternatively, the conversion tract could be generated by repair of a double-stranded DNA gap with a region of heteroduplex adjacent to the repair tract (*Figure 8E*). The Class 4 CO events resemble the classic DSBR pathway except heteroduplexes are

**Table 1.** Description of classes of 'simple' recombination events and number of observed events in each class.[1]

| Class | Description of class | Example | Number observed |
|---|---|---|---|
| 1 | NCO, unidirectional heteroduplex, no strand switch; one-ended SDSA, *Figure 1A* | Chromatid 2 of *Figure 7A* | 26 |
| 2 | NCO, single continuous conversion tract on one side of DSB site | Chromatid 3 of *Supplementary file 2* - S4 | 18 |
| 3 | NCO, hybrid heteroduplex/conversion tract located on one side of DSB site | Chromatid 4 of *Supplementary file 2* - S5 | 12 |
| 4 | CO, heteroduplex on only one chromatid on one side of DSB site | Chromatids 1 and 4 of *Supplementary file 2* - S21 | 5 |
| 5 | NCO involving only one chromatid with heteroduplex on one side of DSB and conversion tract on the other | Chromatid 3 of *Supplementary file 2* - S14 | 4 |
| 6 | CO, uni-directional heteroduplexes on both chromatids propagated in opposite directions | Chromatids 1 and 3 of *Supplementary file 2* - S16 | 2 |
| 7 | CO, no detected heteroduplex or conversion tract on either chromatid | Chromatids 1 and 3 of *Supplementary file 2* - S24 | 2 |
| 8 | NCO, one chromatid with uni-directional conversion tract and one chromatid with uni-directional heteroduplex on same side of DSB | Chromatids 2 and 3 of *Supplementary file 2* - S67 | 2 |

[1]These classes are further described in the text and are depicted in *Figure 8*.

 

observed on only one site of the putative DSB site. One possibility is that the event occurs by the DSBR pathway but one heteroduplex does not include a heterozygous SNP and, therefore, is not observable (*Figure 8F*). Studies of meiotic recombination suggest that heteroduplexes flanking the recombination-initiating DNA lesion are often of different lengths (*Merker et al., 2003*; *Jessop et al., 2005*).

Class 5 NCO events are explicable as reflecting the repair of one broken end with a double stranded DNA gap and one broken end with the canonical processing of one strand (*Figure 8G*) Following SDSA, a recombinant chromatid in which the putative DSB site is flanked by a heteroduplex on one side and a conversion tract on the other will be produced. The Class 6 CO events have the structure predicted for a CO in the DSBR pathway (*Figure 1*, *Figure 8H*). There were only two such events in our dataset. Class seven events resemble Class four and Class six events except there are no observed heteroduplexes on either side of the putative DSB site. Class 7 COs might reflect intermediates with short heteroduplexes or heteroduplexes that are in regions that do not include SNPs (*Figure 8I*).

In the Class 8 NCO events, both interacting chromatids have recombinant SNPs, and the conversion tract on one chromatid overlaps with the heteroduplex tract on the other (*Figure 8J and K*). In *Figure 8J*, this class is generated by formation of a dHJ with one region of short/undetectable heteroduplex, followed by resolution of the dHJ as a NCO. An alternative possibility is that branch migration occurs to generate regions of heteroduplex on both chromatids (*Figure 8K*). Mlh1-independent repair of mismatches in one, but not both, of the heteroduplexes could produce the observed pattern.

If we assume that not all heteroduplexes are detectable, Classes 1, 4, 6, and seven are consistent with the DSBR model of *Figure 1*. For CO classes with uni-directional heteroduplexes on both chromatids (propagated in opposite directions), all events had the pattern expected for nick-directed resolution of Holliday junctions. As shown in *Figure 1*, nick-directed cleavage of the junctions (sites marked as 1, 2, 7 and 8) result in a different pattern of heteroduplexes than cleavage of unnicked junctions (sites marked as 2, 4, 5, and 6). Previously, a bias of the same type was detected for recombination events involving plasmid integration (*Mitchel et al., 2010*). This observation is also consistent with previous studies that showed that *mus81* strains, but not *yen1* strains, had reduced frequencies of crossovers without an effect on gene conversions (*Ho et al.,* 2010; *Yin and Petes, 2015*); Mus81, but not Yen1, has a substrate preference for nicked junctions (*Schwartz and Heyer, 2011*).

Most of the other events require Mlh1-independent mismatch correction or the repair of a double-stranded DNA gap. These classes, however, represent only half of the total 'simple' events. As discussed below, the complex classes require even more radical departures from the canonical recombination model.

## Complex classes of CO and NCO chromatids

Those events that were classified as complex are shown in *Supplementary file 1* - Table S3. A more detailed description of the complex classes is in *Supplementary file 1* - Table S4, and possible interpretations of these events are shown schematically in *Supplementary file 2*. Of the SCB events, 35 of 40 were 'simple' (Classes 1–8) and only five were complex. Assuming the same distribution for the 38 DSCB events, the expected frequencies of DSCBs with two simple events, one simple and one complex event, and two complex events are 29, 8, and 1, respectively. We observed (*Supplementary file 1* - Table S2) 12, 11, and 15 of these classes, respectively, a very significant (p<0.0001 by chi-square test) departure from this expectation. This observation suggests that the timing of the recombinogenic DSB ($G_1$ versus $G_2$) may influence the complexity of its repair. Below, we will discuss two of these complex events in detail. We note that many of these events can be explained by more than one mechanistic pathway, and we have tried to describe the simplest one.

The event depicted in *Supplementary file 2* - S8 (re-drawn in *Figure 9A*) is a DSCB event initiated on the red chromosome. The CO occurred between chromatids 1 and 4. One feature of the CO that is not consistent with the canonical DSBR model is that heteroduplexes are observed at the same position on both crossover chromatids. This pattern is consistent with the possibility of branch migration of one of the Holiday junctions in the rightward direction, resulting in a region of symmetric heteroduplexes (*Figure 9B*). Since the heteroduplex tract is longer on chromatid four than on

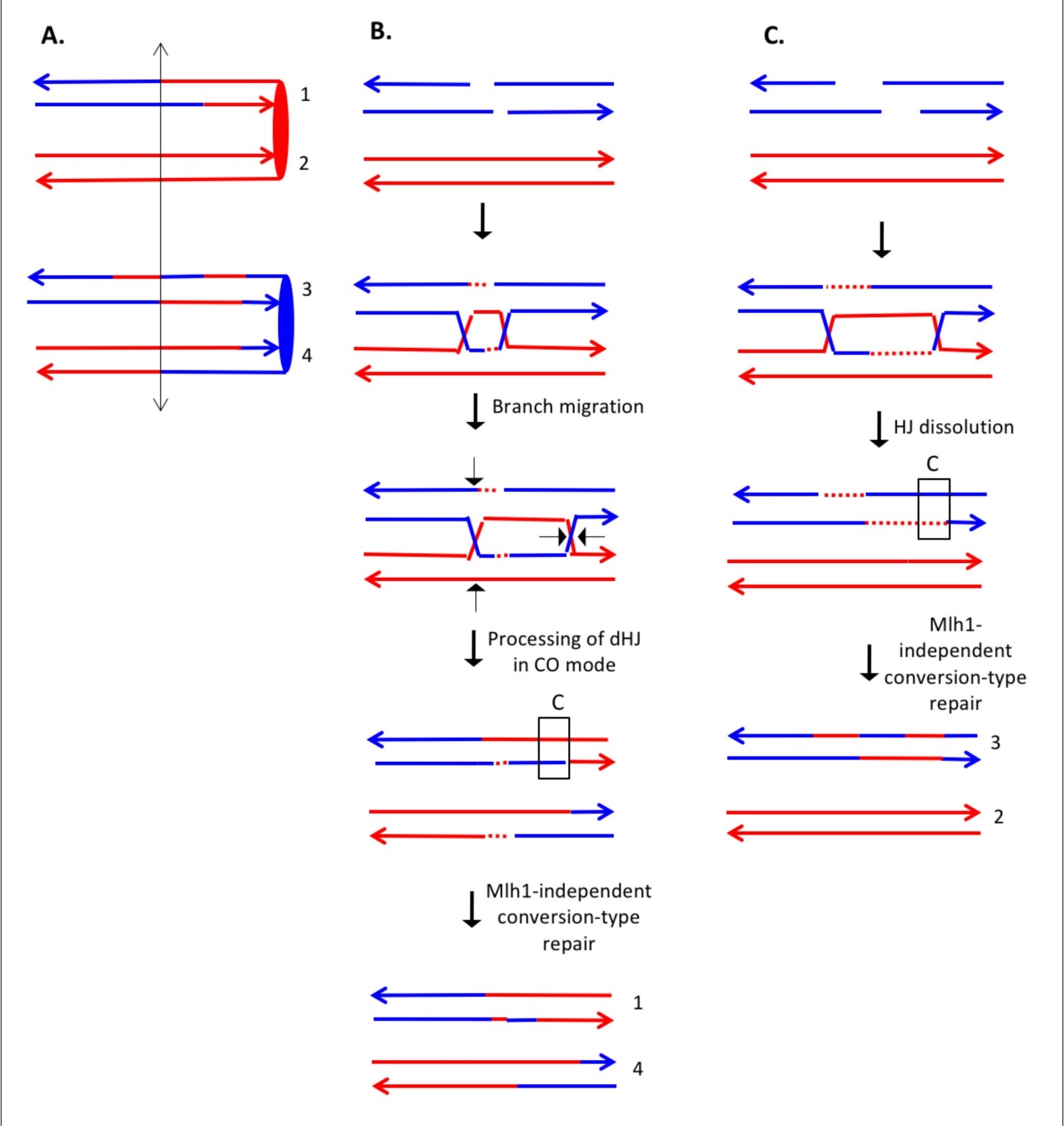

**Figure 9.** Complex DSCB event derived from analysis shown in *Supplementary file 2 - S8*. We infer that the recombination-initiating DSB was on a blue chromosome that was replicated to produce two broken blue chromatids. (A) Depiction of heteroduplexes and conversion tracts present in the mother cell before chromosome segregation. The double-headed arrow indicates the position of the initiating DSB. (B) Mechanism to produce the CO chromatids 1 and 4. Following invasion of the left end of the broken chromatid, one HJ underwent branch migration resulting in a region of symmetric heteroduplexes. The dHJ intermediate was processed to yield a CO as shown by the short arrows. Mismatches in a region of the heteroduplex on the upper chromatid were repaired to yield a conversion event. We suggest that the small homoduplex region near the DSB site did not include a SNP and was, therefore, undetectable. (C) Mechanism to produce the NCO chromatids 2 and 3. Following dHJ formation, the intermediate was dissolved,
*Figure 9 continued on next page*

*Figure 9 continued*

resulting in bidirectional heteroduplexes on chromatid 2. In addition, as in *Figure 8B*, we hypothesize that there was a region of mismatch repair in one of the heteroduplexes.

chromatid 1, there was likely a region of Mlh1-independent conversion-type repair of mismatches on chromatid 4. In the NCO event (involving chromatids 2 and 3), chromatid two had two regions of heteroduplex in which there was a strand switch at the junction of the putative DSB site. This pattern can be explained by formation of a dHJ that was resolved by dissolution (*Figure 9C*). As for the CO event, we also need to postulate a region of Mlh1-independent MMR at one end of the heteroduplex tract.

The event shown in *Supplementary file 2* - S16 (re-drawn in *Figure 10A*) is also a DSCB event with the CO involving chromatids 1 and 3, and the NCO involving chromatid 4. The CO event is consistent with the classic DSBR model (*Figure 10B*). If we assume that the NCO event was initiated by a DSB at the same position, the heteroduplex spanning the DSB site is inconsistent with the DSBR model. One mechanism to explain the observed NCO pattern is that the right broken end undergoes degradation of both strands, and the left end is extended by invasion of the sister chromatid (*Figure 10C*). This invading end is then unpaired from the sister chromatid and pairs with the right end. After processing the dHJ in the NCO mode, the net result of these steps is a recombinant chromatid with a region of heteroduplex that spans the initiating DSB site.

Of the 46 complex events described in *Supplementary file 1* - Tables S3 and S4, about 40% (18 of 46) can be explained by mechanisms involving template switching, branch migration, or independent invasion of two broken ends. In addition, this relatively large fraction of complex events is not solely a characteristic of UV-induced events in *mlh1* strains. About onethird of the spontaneous events in a wild-type strain (*St Charles and Petes, 2013*), and 15% of the UV-induced events in a wild-type strain (*Yin and Petes, 2013*) appear associated with 'patchy' repair of mismatches and/or branch migration.

## Lengths of conversion/heteroduplex tracts in UV-treated YYy310

The median lengths of conversion/heteroduplex tracts in NCO and CO events in UV-treated YYy310 were 5.4 kb (95% confidence limits [CL] of 3.6–7.2 kb) and 10 kb (CL 5.7–17 kb), respectively; the median length of all tracts was 6.1 kb (CL 4.8–8.5 kb). For the CO events, we summed the lengths of the tracts on the two CO chromatids, since these events reflect the repair of a single DSB. These lengths are similar to those previously determined for the UV-treated isogenic wild-type strain (*Yin and Petes, 2013*): 5.7 kb (CL 4.5–6.6) for NCO events; 8.2 kb (CL 6.6–10.3) for CO events; 6.4 kb (CL 5.8–7.3) for all tracts. Thus, the Mlh1p does not have a significant role in determining the length of conversion tracts (*Supplementary file 1* - Table S7).

## Analysis of spontaneous crossovers on chromosome V in YYy310

Although most of the conclusions from our study are based on UV-induced events in YYy310, we also examined five spontaneous crossovers on chromosome V by similar procedures. No unselected crossover or BIR events were observed among these sectors. The coordinates for LOH transitions in the granddaughter cells of each colony, and the classification of these events as Classes 1–8 or complex events are shown in *Supplementary file 1* - Tables S2 and S3, respectively. The events are depicted in *Supplementary file 2* - S81-S85, and the mechanisms that explain these events are in *Supplementary file 2* - S125-S129.

All five sectored colonies reflected DSCBs with one CO and one NCO event. Nine of the ten COs and NCOs were complex with similar features to the UV-induced complex events (Mlh1-independent repair, gap repair, branch migration, and independent invasion of two broken ends) (*Supplementary file 1* - Table S4). Thus, the complexity of the patterns of recombination observed in the UV-induced events is not likely to be a consequence of the multiple DNA lesions introduced by UV. The higher proportion of complex events relative to the UV-induced exchanges may be a consequence of the nature of the recombinogenic lesion or differences in the mechanism of DNA repair in cells that have a single DNA lesion versus those with high levels of DNA damage.

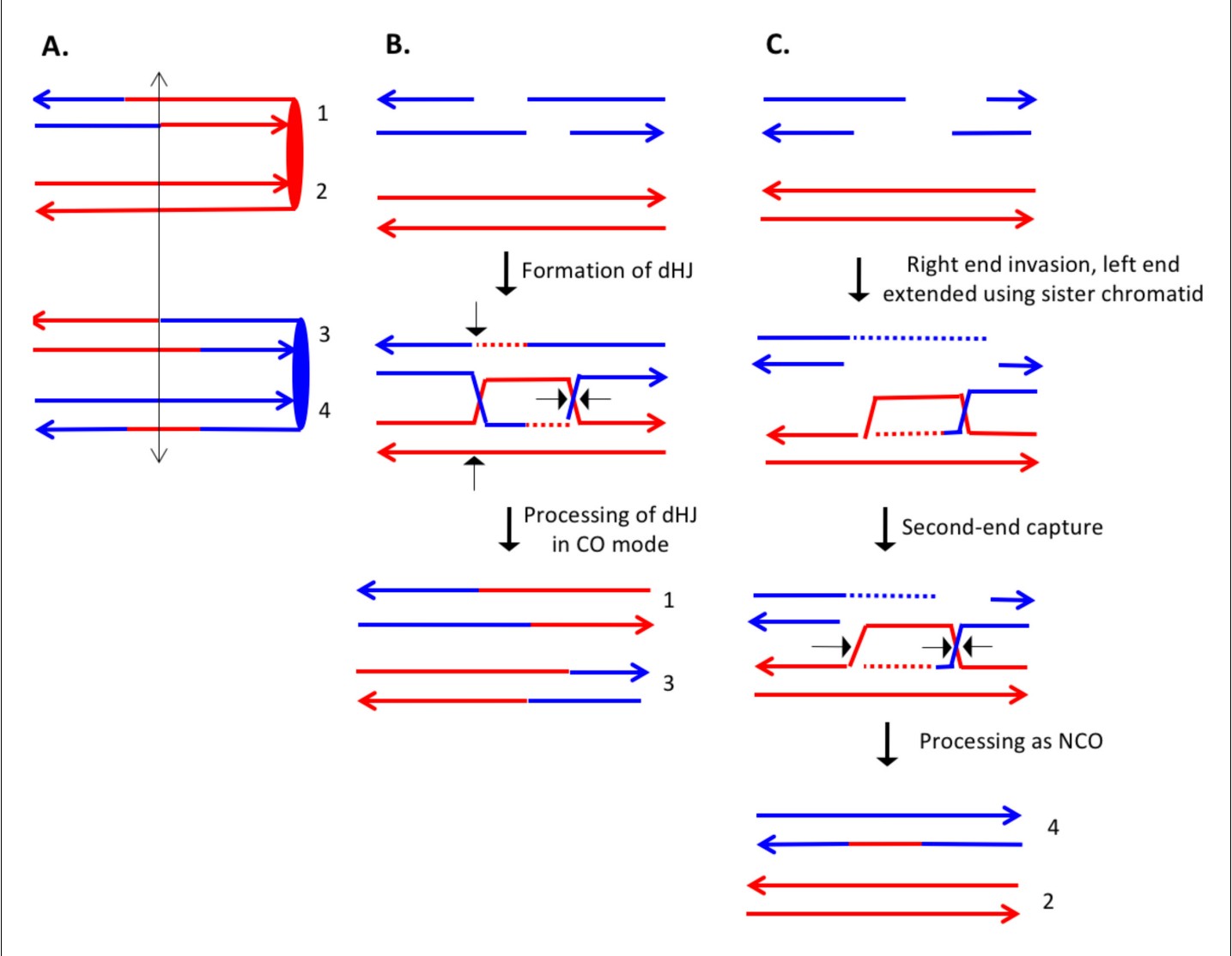

**Figure 10.** Complex DSCB event derived from the analysis shown in *Supplementary file 2* - S16. As in *Figure 8*, the initiating DSB was on the blue chromosome. (**A**) Depiction of recombinant chromatids in mother cell before segregation of chromosomes into the daughters. (**B**) CO between chromatids 1 and 3. The pattern of heteroduplexes on these chromatids is that expected from the DSBR model. (**C**) NCO between chromatids 2 and 4. In this event, the heteroduplex spans the putative DSB site on chromatid 4. The right broken end underwent resection of both strands before strand invasion. The left end was extended by invasion of the sister chromatid. Following annealing of the extended end to the D-loop, the intermediate was processed as a NCO. We suggest that the small region of heteroduplex on the red chromatid did not contain a SNP and was not detectable.

## Analysis of spontaneous crossovers at the HS4 hotspot on chromosome IV in YYy311

Previously, we showed that an inverted pair of Ty elements resulted in a spontaneous mitotic recombination hotspot (termed 'HS4') on the right arm of chromosome IV (*St Charles and Petes, 2013*). We subsequently showed that HS4-associated conversion events were associated with very long (>25 kb) conversion tracts (*Yim et al., 2014*); conversion tracts associated with a similar pair of inverted Ty elements on chromosome III also have very long conversion tracts (median length of 41 kb; *Chumki et al., 2016*). To monitor the activity of this hotspot in an *mlh1* strain, we inserted the *hphMX4* and *URA3* markers centromere-proximal and centromere-distal to HS4 on one of the two homologs in the *mlh1* diploid YYy311. In addition, the YJM789-derived copy of IV has an insertion of *SUP4-o* near the right telomere. A crossover between the two markers results in a red/white

sectored colony in which the white sector is resistant to hygromycin (Hyg$^R$) and 5-fluoro-orotate (5-FOA$^R$), and the red sector is Hyg$^R$ 5-FOA$^S$ (*Figure 11*). In an isogenic wild-type strain, 17% of sectored red/white colonies result from a crossover in the *hphMX4-URA3* interval (*St Charles and Petes, 2013*). 17% (101/593) of the spontaneous sectoring events in the isogenic *mlh1* strain YYy311 occurred in this same interval, indicating that loss of Mlh1 does not reduce the activity of HS4. We examined six of the sectored colonies by SNP microarrays (*Supplementary file 1* - Tables S2-S4), and five had patterns of recombination indicating an HS4-initiated event (shown in *Supplementary file 2* - S87-S91). Since the recombination event in sectored colony shown in *Supplementary file 2* - S86 is initiated somewhere in the interval 968–978 kb, and HS4 is located in the interval 981–993 kb, this event is not likely to be HS4-initiated. Since the 56 kb *hphMX4-URA3* interval is considerably larger than the size of the HS4 hotspot (12 kb), it is not surprising that some crossovers in the *hphMX4-URA3* region are not HS4-mediated. Mechanisms consistent with the HS4-mediated events are shown in *Supplementary file 2* - S87-S90.

Based on the five HS4-initiated events, we can make several generalizations. In all events, at least three chromatids had transitions between W303-1A- and YJM789-derived sequences, indicating that all HS4-initiated events involved the repair of two sister chromatids broken at approximately the same position. Second, all events are initiated on the red (W303-1A-derived homolog), as expected since the YJM789-derived homolog lacks HS4 hotspot activity (*St Charles and Petes, 2013*). Third, with the exception of the event in *Supplementary file 2* - S88, gaps of at least 25 kb are observed in the HS4-initiated events; these gaps are much larger than necessary to remove the approximately 9 kb HS4-related heterology that is present in W303-1A but absent in YJM789-derived homolog (*St Charles and Petes, 2013*). Fourth, many of these gaps are flanked by heteroduplexes on one or both sides. Gaps can be propagated symmetrically from HS4 (*Supplementary file 2* - S89) or with a bias toward or away from the centromere. Fifth, some of the gaps (regions of 4:0 segregation without heteroduplexes) are interrupted by regions of heteroduplex (chromatid 4 of *Supplementary file 2* - S91). Such a pattern can be explained by a template switch to a sister chromatid as shown in *Supplementary file 2* - S91.

We also used SNP microarrays to examine twelve red/white sectored colonies with a crossover in the *hphMX4-URA3* interval in UV-treated YYy311. Since it was clear that none of these twelve colonies had an HS4-mediated event, we did not attempt to look for two genotypes within each sector, In addition, a smaller fraction of events (10%, 10/102) were in the *hphMX4-URA3* interval in the UV-treated samples of YYy311 than in the untreated samples. Thus, HS4 is a hotspot for spontaneous, but not UV-stimulated recombination events.

## Effect of the *mlh1* mutation on the frequency of mitotic crossovers

Two different assays can be used to monitor the effect of the *mlh1* mutation on crossovers: the frequencies of spontaneous and UV-induced red/white sectored colonies compared to wild type, and the frequency of unselected crossovers in UV-treated cells. These two assays yield somewhat different results. We observed slightly more (<two-fold) spontaneous red/white sectored colonies in YYy311 than in the isogenic wild-type strain (53 sectored colonies/998395 total colonies [$5.3 \times 10^{-5}$/division]) in the *mlh1* strain versus 55 sectors/1761664 total [$3.1 \times 10^{-5}$/division; *St Charles and Petes (2013)*. A similar slight elevation was observed for UV-treated YYy311 cells: 378 sectors/3228 total (11.7%) in the *mlh1* strain versus 127 sectors/1420 total (8.9%) in the isogenic wild-type strain. In addition, the frequency of UV-stimulated sectored colonies in YYy310 (*SUP4-o* on chromosome V) is higher than the UV-stimulated events in the isogenic wild-type strain: 45 sectors/2980 colonies (1.5%) and 68 sectors/7194 colonies (0.9%), respectively.

An elevated frequency of crossovers could be explained by the heteroduplex-rejecting properties of Mlh1p (*Harfe and Jinks-Robertson, 2000*). If this mechanism was relevant to our system, we might expect a higher density of SNPs and in/dels in the heteroduplex tracts of *mlh1* strains than observed in tracts in the wild-type strain. In the UV-treated *mlh1* strain, we observed 2688 SNP+in/dels in tracts with a total length of 635041 bp, a density of 0.0042. In the UV-treated wild-type strain (*Yin and Petes, 2013*), we found 15387 SNP+in/dels in tracts totaling 4573874 bp, a density of 0.0034. By a Mann-Whitney comparison, the difference in density is of borderline significance (p=0.06).

In contrast to the first assay, the second assay indicates that Mlh1p positively influences the frequency of crossovers. In our previous study of unselected recombination events, in twenty UV-

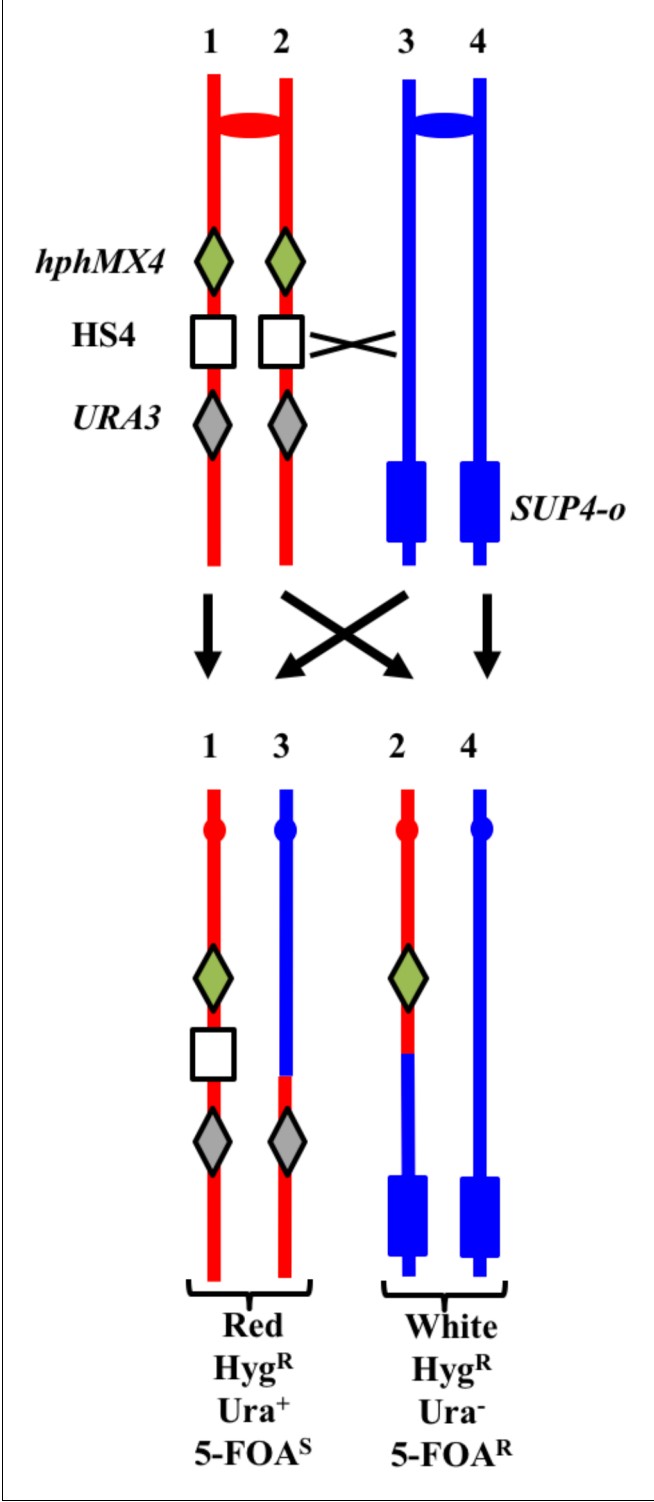

**Figure 11.** System for detecting crossovers at the HS4 hotspot on chromosome IV (strain YYy311). The W303-1A- and YJM789-derived chromatids are shown in red and blue, respectively. On the W303-1A-derived homolog, the HS4 hotspot is flanked by the *hphMX4* and *URA3* markers; *SUP4-o* is located near the telomere of the YJM789- derived homolog. Since the recombinogenic DSB is located at HS4, HS4-initiated events usually result in loss of HS4 as shown.

induced sectored colonies in the wild-type strain (*Yin and Petes, 2013*), we observed 141 interstitial LOH events, 50 COs, and 10 BIR events for a total of 201 LOH events (10.1/sectored colony). In seven sectored colonies of the *mlh1* YYy310 strain, we found 65 interstitial LOH events, 6 COs, and 0 BIR events for a total of 71 LOH events (10.1/sectored colony) (*Supplementary file 1* - Table S8). The reduction in the number of COs is statistically significant (p=0.003 by Fisher exact test). Although the discrepancy in the two assays prevents a strong conclusion about the effect of Mlh1p on mitotic crossovers, the direct assay of unselected crossovers by microarrays is less prone to artifacts than the frequency of red/white sectors since colony color can be affected by a variety of different types of genetic alterations including mitochondrial mutations (*Kim et al., 2002*).

## Discussion

The main conclusion from our analysis is that about one-third of mitotic recombination events are not explicable by the simplest forms of the double-strand-break repair model. More specifically, we need to invoke branch migration of Holliday junctions, template switching during repair synthesis, repair of double-stranded DNA gaps, and/or Mlh1-independent MMR. Our analysis confirms our previous conclusion that more than half of spontaneous crossovers and recombination events induced by high doses of UV reflect the repair of two sister chromatids broken at approximately the same position, likely resulting from replication of chromosome broken in $G_1$. We also show that mitotic recombination events involve heteroduplexes that are longer than those observed for meiotic exchanges. In a previous study, we characterized a hotspot for spontaneous mitotic recombination events (HS4) associated with an inverted pair of Ty elements (*St Charles and Petes, 2013*). In the current study, we show that the very long conversion tracts associated with HS4-initiated events (*Yim et al., 2014*) are associated with the repair of long double-stranded DNA gaps. In addition, HS4 hotspot activity is independent of Mlh1p and is not induced by UV.

### Complexity of recombination events

Before discussing exceptions to the recombination model shown in *Figure 1*, it is important to point out that about two-thirds of the observed events <u>do</u> fit this model. Among the exceptional events are those in which the two interacting chromatids have heteroduplexes involving SNPs at the same positions (for example, the CO chromatids 1 and 4 in *Supplementary file 2* - S8). Symmetric heteroduplexes of this type can be explained as a consequence of branch migration of a Holliday junction. We previously suggested that branch migration may explain complex patterns of SNPs in MMR-proficient strains (*St Charles et al., 2012*; *Yin and Petes, 2013*). Branch migration during meiotic recombination in yeast would be detectable as two spores with post-meiotic segregation within one tetrad (aberrant 4:4). Such events are rare in yeast, although a few percent of the total aberrant segregants had this pattern in an *msh2* strain (*Martini et al., 2011*).

In the current DSBR model, gene conversion events are a consequence of the repair of mismatches in heteroduplexes. Thus, in the absence of MMR, conversion events should be absent. In our study, we frequently observed conversion tracts, some adjacent to a heteroduplex tract (*Supplementary file 2* - S5, chromatid 4) and some 'solo' conversions (*Supplementary file 2* - S4, chromatid 3) One possible source for such events is Mlh1-independent MMR. In cell-free extracts, MMR is only partially defective in the absence of Mlh1 if there is a 5' nick near the mismatch (*Genschel and Modrich, 2009*). In addition, *Coïc et al. (2000)* identified a short-patch repair system in *S. cerevisiae* that was independent of the classic MMR system. The exonuclease activity of DNA polymerase δ could also remove mismatches close to the DSB (*Anand et al., 2017*). If these uncharacterized systems are responsible for the conversion events observed in our studies, they are most likely to be involved in those events with short conversion tracts. We suggest that long conversion tracts (chromatid 2, *Supplementary file 2* - S33; *Supplementary file 2* - S87-91) likely reflect the repair of a large double-stranded DNA gap. Although DSBs are usually thought to be processed by 5'−3' degradation of the broken ends, resection of the 3' 'tails' has also been observed (*Zierhut and Diffley, 2008*). It is clear that *S. cerevisiae* has the ability to repair double- stranded DNA gaps (*Orr-Weaver et al., 1981*) and, in the first version of the DSBR model, both gap repair and MMR were presented as mechanisms for generating a gene conversion event (*Szostak et al., 1983*).

We also observed both short and long tracts of restoration. For example, the long heteroduplex tract in chromatid 2 of *Supplementary file 2* - S3 is interrupted by several one-SNP restoration tracts; such tracts may reflect Mlh1-independent MMR. In *Supplementary file 2* - S44, a long heteroduplex tract on chromatid one is interrupted by a long tract of restoration. As shown in *Supplementary file 2* - S44, this pattern can be explained by a template switch, followed by SDSA. In invoking template switching to explain certain patterns of SNP segregation, we point out that such switches were also proposed as intermediates in the production of complex meiotic recombination events (*Martini et al., 2011*). In addition, template switching is commonly observed during break-induced replication (*Smith et al., 2007*). Since the sister chromatid is a preferred substrate for the repair of DSBs in mitosis (*Kadyk and Hartwell, 1992*; *Bzymek et al., 2010*), a template switch to the sister chromatid would not be unexpected.

In previous mitotic (*Miura et al., 2012*) and meiotic recombination studies (*Martini et al., 2011*), some events appear to reflect independent interactions of the left and right broken ends with the intact template. For example, Martini *et al.* proposed that a double SDSA event could produce a NCO chromatid in which heteroduplexes with a strand switch were separated by a region of restoration. Similarly, in our analysis, some events are likely to reflect double-ended invasions. In *Supplementary file 2* - S83, the CO chromatid four has a heteroduplex with a strand switch that flanks the putative DSB site. This pattern, which is unexpected by the DSBR model, can be explained by invasion of the right broken end, followed by processing, and subsequent invasion of the left broken end.

The complex patterns of recombination observed in our study are also found in global analysis of meiotic recombination events in wild-type (*Mancera et al., 2008*) and *msh2* (*Martini et al., 2011*) strains. In the Mancera *et al.* study, more than 10% of the conversion tracts were described as 'complex', and, in the Martini *et al.* analysis, the majority of the CO associated heteroduplex tracts required Msh2-independent MMR, branch migration, template switches, and/or the repair of double-stranded gaps. These observations contrast with other studies of mitotic recombination events induced with an endonuclease at a specific site; such events tend to have simpler patterns of marker segregation (*Nickoloff et al., 1999*; *Mitchel et al., 2010*). Our studies differ from most previous studies in several ways: the nature of the recombinogenic lesions, the timing of the DSBs during the cell cycle, and the diverse genetic location of the recombinogenic lesions. With regard to the third point, if DNA lesions at different regions of the genome utilize different pathways of repair, as suggested by meiotic recombination studies (*Medhi et al., 2016*), then a global analysis of recombination would reveal more heterogeneity of recombinant products than studies based on a single locus. Lastly, we point out our analysis is performed in diploid cells whereas many of studies of HO-induced events are done in haploids with duplicated sequences. Any or all of these factors may result in the greater complexity observed in our recombination events.

## Comparison of recombination events in wild-type and *mlh1* strains

In general, our observations support the main conclusions of our previous studies of spontaneous (*St Charles and Petes, 2013*) and UV-induced (*Yin and Petes, 2013*) mitotic recombination. In wild-type and *mlh1* strains, exchanges induced by high levels of UV in $G_1$-synchronized cells represent DSCBs and SCBs with approximately the same frequency. All five of the spontaneous events selected on chromosome IV in the *mlh1* strain YYy311 were DSCB events as were the majority of spontaneous events in the wild-type strain (*St Charles and Petes, 2013*). In contrast, recombination events induced by low levels of DNA polymerase alpha (*Song et al., 2014*) or delta (*Zheng et al., 2016*) usually reflect the break of a single chromatid.

*Esposito (1978)* previously proposed that mitotic recombination in yeast was initiated in $G_1$-associated DSB that formed a Holliday junction connecting the two unreplicated homologs. Mismatch correction occurred in the heteroduplex regions, and the resulting intermediate was replicated to produce both CO and NCO recombinant products. By this model, granddaughter cells would not be expected to be genotypically different. Our analysis rules out this model.

In addition to its role in the repair of mismatches, the Mlh1p has two other functions related to recombination. First, in conjunction of Mlh3p, Mlh1p stimulates meiotic crossovers (*Hunter and Borts, 1997*; *Wang et al., 1999*); this function is likely related to its single-strand nicking activity (*Rogacheva et al., 2014*). If Mlh1p had a similar role in mitotic recombination, we would expect a hypo-Rec phenotype in *mlh1* strains. Conversely, for recombination substrates with diverged

sequences, Mlh1 has anti-recombination activity (*Nicholson et al., 2000*). Strains lacking *mlh1* have seven-fold more crossovers than wild-type strains for substrates with 9% sequence divergence. Although this degree of divergence is much greater than the average 0.5% divergence existing between the homologs in YYy310 and YYy311, strains lacking the Msh2 and Msh3 repair proteins have an elevated frequency of crossovers relative to wild-type for 350 bp substrates with a single mismatched base (*Datta et al., 1997*).

Based on the data described above, it is difficult to predict the effect of the *mlh1* mutation on mitotic recombination in our system. As discussed in the Results section, we find an elevated frequency of sectored colonies in the *mlh1* strain compared to the wild-type strain, but a reduced frequency of crossovers among unselected UV-induced recombination events. Although the results indicating a several-fold reduced frequency of crossovers in the *mlh1* strain are likely to be less prone to artifacts than the sectored colony assay, further experiments are needed to substantiate them.

## Analysis of the HS4 mitotic recombination hotspot

Previously, we demonstrated that an inverted pair of Ty elements on the W303-1A-derived copy of chromosome IV stimulated mitotic recombination (*St Charles and Petes, 2013*); this hotspot was called 'HS4'. The HS4-associated recombination events have the following properties (*St Charles and Petes, 2013*; *Yin and Petes, 2014*; *Yim et al., 2014*): 1) they are DSCB events, 2) both of the Ty elements are required for hotspot activity, 3) the events are initiated on the W303-1A homolog presumably because the YJM789 homolog lacks the inverted pair of elements, 4) HS4 activity requires the Exo1p, and 5) conversion events associated with HS4 are extraordinarily long with a median size >50 kb. Our analysis of five HS4-related events (*Supplementary file 2* - S87-91) shows that most have the properties expected for the repair of a large double-stranded gap (long conversion events) with flanking heteroduplex regions. Since the HS4 hotspot on the W303-1A-derived homolog is a pair of inverted repeats and the comparable region on the YJM789 homolog has only a partial Ty element in the same location (*St Charles and Petes, 2013*), a DSB occurring within HS4 would require a region of end resection of at least 10 kb to expose sequence homologies between the two homologs. In general, the conversion/heteroduplex tracts for the HS4-induced events exceed 10 kb by a considerable amount (*Supplementary file 1* – Table S9). For example, in the NCO event on chromatid 2 of *Supplementary file 2* - S91, the total length of the conversion/heteroduplex region is 39 kb, comparable to the lengths of conversion tracts observed in the wild-type strain (*Yim et al., 2014*).

Our data show that the very long conversion tracts observed in wild-type strains are not a consequence of extremely long heteroduplexes, but reflect an intermediate that does not have mismatched bases. Since we observe many other events that are explicable as gap repair, this is our preferred explanation for the HS4-associated events, which is consistent with the observation that the activity of HS4 requires Exo1p (*Yin and Petes, 2014*). The possible role of Exo1p is the expansion of a single-stranded nick into a large single-stranded gap before DNA replication, facilitating the formation of secondary structures involving the inverted Ty elements that could be subsequently processed by structure-specific nucleases to form the recombinogenic DSB. As described above, the broken ends would require extensive single-stranded processing because of the heterology in order to invade the homolog. These long single-stranded regions could be susceptible to endonucleolytic attack, resulting in the observed double-stranded gaps. We cannot, however, rule out the generation of these long conversion tracts by a BIR-like mechanism (*Yim et al., 2014*). We found that HS4-associated hotspot was not stimulated by UV. This result argues that the activity of HS4 is not a consequence of random DNA breaks located near in the inverted repeat, but is likely a consequence of a DSB induced in a secondary structure in the DNA that is formed independently of nearby DSBs. Lastly, we suggest that the long conversion tracts associated with HS4 might be a general property of recombination events initiated in a large heterozygous insertion.

## Summary

In conclusion, our global analysis of UV-induced and spontaneous mitotic recombination events in an MMR-deficient yeast strain demonstrates that many of the events are more complex than predicted

by the current DSBR model. We find that many gene conversion tracts are generated by a mechanism that does not involve the Mlh1-dependent repair of mismatches within a heteroduplex.

## Materials and methods

### Strains

Constructions of the hybrid diploid strains YYy310 (isogenic isolates YYy310.9, YYy310.10) and YYy311 (isogenic isolates YYy311.1 and YYy311.3) are described in detail in *Supplementary file 1* - Table S1. In brief, the *MLH1* gene was deleted from haploid derivatives of W303-1A and YJM789; these strains have about 0.5% sequence divergence (*Wei et al., 2007*). The primary difference between YYy310 and YYy311 is the location of the tRNA suppressor gene *SUP4-o*: located near the left telomere of the YJM789-derived chromosome V homolog in YYy310, and near the right telomere of the YJM789-derived chromosome IV homolog in YYy311. As explained in the text, this heterozygous insertion allows detection of crossovers as red/white sectored colonies. In addition, in the YYy311 strain on the W303-1A-derived homolog, the previously-described HS4 hotspot (*St Charles and Petes, 2013*) was flanked by *hphMX4* and *URA3* markers. In both YYy310 and YYy311, the *MATα* locus was deleted to allow G$_1$-synchronization with α-factor. This deletion also prevents the strains from undergoing meiosis in normal growth conditions. As noted below, however, such strains can be induced to undergo meiosis using special sporulation conditions.

### Isolation of granddaughter cells derived from red/white sectors

Most of the details of this analysis are described in the main text. In brief, we purified about 10–20 individual colonies from the red and white sectors. We analyzed one purified colony by SNP-specific microarrays to determine the approximate location of the selected recombination event. We then identified SNPs within these regions that could be assayed by a combination of PCR followed by treatment of the resulting PCR fragment with a diagnostic restriction enzyme. This procedure was previously used to detect LOH events on chromosome V (*Lee et al., 2009*). We employed this type of analysis with about 10–20 individual red and white colonies derived from each sector using the PCR primers and restriction enzymes listed in *Supplementary file 1* - Table S5. If two PCR samples produced different fragment sizes after treatment with the restriction enzyme, we concluded that these strains represented the two granddaughters within the sector. We were able to detect most granddaughter cells using SNPs located on chromosomes V (YYy310) or IV (YYy311). However, for some sectored colonies, we used SNPs located on other chromosomes (*Supplementary file 1* - Table S5). For some chromosomal regions, the SNPs did not result in an altered restriction enzyme recognition site. Consequently, for some events, we examined SNPs by sequencing PCR fragments from the relevant regions of the genome. The primers used in this analysis are also shown in *Supplementary file 1* - Table S5.

### Meiotic analysis of marker coupling

For granddaughter cells with multiple heterozygous regions interspersed with homozygous regions (for example, *Figure 6A*), we determined the coupling of these regions by examining microarrays of single spore cultures derived from the four granddaughters. Although the diploidsYYy310 and YYy311 lack *MATα* locus, these strains can be sporulated in medium containing 5 mM nicotinamide (*St Charles et al., 2012*). By examining a single spore, we could generally determine the coupling relationships of the markers in each granddaughter. For example, the pattern of hybridization to SNPs shown for the spore in *Figure 6* 8 demonstrates that the two regions of heteroduplex in the granddaughter cell are on different homologs. For most of the granddaughter cells, the SNP patterns of one daughter spore allowed unambiguous conclusions. However, if the pattern of SNPs in the spore was inconsistent with that observed in the diploid or if additional crossovers were observed within 50 kb of the mitotic recombination event, we considered the possibility of a meiotic crossover. Consequently, for such events, we examined the segregation pattern of SNPs flanking the region of mitotic exchange in several additional spores derived from the granddaughter cells. The primer pairs and restriction enzymes used in this analysis are described in *Supplementary file 1* - Table S5. Complicating meiotic recombination events were rare, as expected, because the *mlh1*

mutation substantially reduces the frequency of meiotic crossovers in yeast (*Hunter and Borts, 1997*).

Spontaneous red/white sectors generally had no unselected events. To determine coupling relationships within heterozygous regions, we used a different procedure from that described for the UV-induced events. We screened for monosomy on the chromosome with the selected event using a marker located centromere-proximal to the exchange. In YYy310, the heterozygous *URA3* gene was on the same chromosome arm (left arm of V) as the selected crossover, but was located centromere-proximal to the mitotic exchange. We selected for loss of the copy of V that had the *URA3* gene by using medium containing 5-fluoro-orotate. We confirmed the loss using PCR analysis, and restriction enzyme digestion of the resulting PCR product. Chromosome V-specific SNP microarrays were then used to determine the coupling of markers along the remaining homolog. A similar procedure was used to examine spontaneous events on yeast chromosome IV in YYy311. In YYy311, we selected monosomic strains that lacked the heterozygous *TRP1* marker located near *CEN4* using medium containing 5-fluoroanthranilic acid (*Toyn et al., 2000*).

## SNP microarrays

We used three types of SNP-Microarrays in our analysis: a whole-genome microarray (*St Charles et al., 2012*) and microarrays to SNPs on chromosome IV (*St Charles and Petes, 2013*) and chromosome V (*Yin and Petes, 2013*). The sequences of the oligonucleotides in the microarrays and their designs are on the Gene Expression Omnibus Website (https://www.ncbi.nlm.nih.gov/geo/) under the addresses GPL20144 (whole-genome), GPL21552 (chromosome IV), and GPL21274 (chromosome V). In brief, for each heterozygous SNP, we designed four 25-base oligonucleotides, two identical to the Watson and Crick strands of the W303-1A allele and two identical to the YJM789 allele. The heterozygous SNP was located near the middle of each oligonucleotide. Since the efficiency of hybridization is higher when genomic DNA is perfectly matched to the oligonucleotide than when there is a mismatch between the genomic DNA and the oligonucleotide, by measuring the ratio of hybridization of the experimental DNA to a control heterozygous strain for the four SNP-specific oligonucleotides, we can determine whether the genomic DNA is heterozygous for SNP or homozygous for either allele. In our experiments, the ratios of hybridization to the different oligonucleotides were normalized by comparing these ratios to a control heterozygous diploid. Genomic DNA from the experimental strains (derived from YYy310 or YYy311) was labeled with the Cy5-dUTP fluorescent nucleotide and DNA from an isogenic wild-type strain was labeled with Cy3-dUTP. The labeled samples were mixed and hybridized to the microarray. Details concerning the hybridization conditions and determining the levels of hybridization to the oligonucleotide probes are in *St Charles et al. (2012)* and *St Charles and Petes (2013)*. The data for all of these hybridization experiments are on the GEO Website (GSE100497).

## Calculation of conversion/heteroduplex tract lengths

Two types of conversion/heteroduplex events were observed in our studies: events unassociated with crossovers and events associated with crossovers. For the first class, we averaged the distance between the closest heterozygous SNPs flanking the tract (the maximum tract length) and the homozygous SNPs located at the borders of the conversion/heteroduplex tract (the minimum tract length). For crossover-associated conversion/heteroduplex tracts, we averaged the distance between the homozygous SNPs flanking the tract (the maximum tract length), and the distance between the SNPs that were within the tract closest to the borders of the crossover.

## Bioinformatics and statistical analysis

The two haploid strains for W303 and YJM789 that were sequenced previously (*St Charles et al., 2012*) were analyzed for the number of SNPs and insertions/deletions (in/dels) in gene conversion tracts. Paired-end reads were aligned to the S288c reference genome version 3 (downloaded from sad Cer3 in UCSC Genome Browser; https://genome.ucsc.edu) by BWAMEM software (*St Charles and Petes, 2013*) and SNPs were determined using samtools (*Li et al., 2009*). Because our SNP-Microarray was based on S288c reference genome version 2 (downloaded from sacCer2 in UCSC Genome Browse), we translated the positions of the SNPs from the microarray to the version three reference genome before counting the number of SNPs and indels within each of the conversion

tracts. We then used BEDTools (*Quinlan and Hall, 2010*) to count the number of SNPs within the conversion tracts. RStudio (http://www.rstudio.com/) used for various statistical tests used in this study.

## Acknowledgements

We thank members of the Petes and Jinks-Robertson labs for suggestions during the course of this work, and Sue Jinks-Robertson, Yee Fang Hum, Dao-Qiong Zheng and Ke Zhang for comments or help with the manuscript. The research was supported by NIH grants GM24110, GM52319, 1R35GM118020, and 5T32GM007754-37.

## Additional information

### Funding

| Funder | Grant reference number | Author |
|---|---|---|
| National Institute of General Medical Sciences | GM24110 | Yi Yin<br>Margaret Dominska<br>Eunice Yim<br>Thomas D Petes |
| National Institute of General Medical Sciences | GM52319 | Yi Yin<br>Margaret Dominska<br>Eunice Yim<br>Thomas D Petes |
| National Institute of General Medical Sciences | 1R35GM118020 | Yi Yin<br>Margaret Dominska<br>Eunice Yim<br>Thomas D Petes |
| National Institute of General Medical Sciences | 5T32GM007754-37 | Yi Yin |

The funders had no role in study design, data collection and interpretation, or the decision to submit the work for publication.

### Author contributions

YY, Conceptualization, Data curation, Formal analysis, Validation, Investigation, Visualization, Methodology, Writing—original draft, Writing—review and editing; MD, EY, Data curation, Investigation; TDP, Conceptualization, Resources, Data curation, Formal analysis, Supervision, Funding acquisition, Visualization, Project administration, Writing—review and editing

### Author ORCIDs

Yi Yin, http://orcid.org/0000-0003-0963-2672
Thomas D Petes, http://orcid.org/0000-0003-4890-8002

## Additional files

### Supplementary files

• Supplementary file 1. Supplementary Tables S1-S9.

• Supplementary file 2. - S1 Mechanisms that generate Classes 1–5 recombination events. For all events, we show the initiating DSB on the blue chromatid. Dotted lines show sequences generated by replication or during mismatch repair. Black arrows indicate the position of the initiating DSB. The mechanisms are discussed in detail in the main text. Red and blue lines represent DNA strands of the W303-1A-derived chromatid and YJM789-derived chromatid, respectively. Arrows on these strands indicate the 3' ends. A. Class 1. Class one events are NCOs formed by one-ended SDSA. B. Class 2. These NCO events could be generated by one-ended SDSA, followed by Mlh1p-independent MMR. C. Class 2. An alterative possibility is that Class two events reflect repair of a double-stranded DNA gap. D. Class 3. In this NCO class, the heteroduplex region is adjacent to a

conversion tract. Such events could reflect a heteroduplex tract in which mismatches are repaired in one part of the tract and left unrepaired in the other. E. Class 3. An alternative model for this class is that the conversion tract is the result of repair of a double-stranded DNA gap with a heteroduplex region at one end. F. Class 4. For this CO class, a heteroduplex is observed on one chromatid but not the other. This class could be explained by the DSBR pathway in which heteroduplex region is short relative to the other; if the short heteroduplex does not contain a mismatch, it would be undetectable. *Supplementary file 2* - S2. Mechanisms that generate Classes 6–8 recombination events. Events are depicted as in *Supplementary file 2* - S1. A. Class 5. In this NCO class, the heteroduplex region is located on the opposite of the DSB site from the conversion region. This pattern is consistent with the repair of a double-stranded DNA gap that was restricted to one of the broken ends. B. Class 6. This CO class is identical to the pattern expected for the DSBR model. C. Class 7. In this CO class, no heteroduplexes or conversion tracts are observed adjacent to the crossover, consistent with the formation of a dHJ with short heteroduplex tracts that do not include mismatches. D. Class 8, mechanism 1. In this NCO class, one chromatid has a conversion tract, and the other chromatid has a heteroduplex involving SNPs at the same position. This event could reflect resolution of a dHJ intermediate in the NCO model in which one region of heteroduplex is undetectable. E. Class 8, mechanism 2. An alternative mechanism involves branch migration of a HJ, followed by resolution of the intermediate in a NCO mode. Mismatches in one of the two chromatids are repaired to generate a conversion tract. *Supplementary file 2* - S3-S91. In the upper part of the figure (labeled A), we show the patterns of SNPs derived from the W303-1A homolog (red) and YJM789 homolog (blue). Each line represents a chromosome. Each pair of lines is derived from one of the granddaughter cells in the white (W1 or W2) or red (R1 or R2) sectors. The locations of the SNPs, represented by colored circles, are drawn proportionally with a scale showing SGD coordinates at the bottom of A. In the B part of the figures, we show the arrangement of SNPs in double-stranded chromatids in the mother cell before chromosome segregation; in this part of the figure, each line represents a single-strand of DNA of the chromatids. The inferred positions of the initiating DSBs are indicated with arrows. Chromatids 1 and 2 are sisters, and chromatids 3 and 4 are sisters. In part C, for recombination events that are classified as 'Complex', we show possible mechanistic pathways for their formation. *Supplementary file 2* - S3-S80 show selected and unselected UV-induced events in strain YYy310. *Supplementary file 2* - S81-S85 show selected spontaneous events on chromosome V in strain YYy310, and *Supplementary file 2* - S86-S91 represent spontaneous HS4-related crossovers on chromosome V in strain YYy311. The coordinates and classifications of the events represented by these supplemental figures are in *Supplementary file 1* - Tables S2 and S3, respectively. Lastly, we note that the event depicted in *Supplementary file 2* - S71 has the pattern expected for a BIR event in one of the daughter cells. Because of this complication, we did not attempt to infer the locations of the initiating DSBs.

## Major datasets

The following dataset was generated:

| Author(s) | Year | Dataset title | Dataset URL | Database, license, and accessibility information |
|---|---|---|---|---|
| Yin Y, Petes TD | 2017 | High-resolution mapping of heteroduplex DNA formed in UV-induced and spontaneous mitotic recombination events in yeast | https://www.ncbi.nlm.nih.gov/geo/query/acc.cgi?acc=GSE100497 | Publicly available at the NCBI Gene Expression Omnibus (accession no: GSE100497) |

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
