## [Decision Letter]

Thank you for submitting your article "High-resolution mapping of heteroduplex DNA formed in UV-induced and spontaneous mitotic recombination events in yeast" for consideration by *eLife*. Your article has been reviewed by three peer reviewers, and the evaluation has been overseen by a Reviewing Editor and Jessica Tyler as the Senior Editor. The following individuals involved in review of your submission have agreed to reveal their identity: Bertrand Llorente (Reviewer #3).

The reviewers have discussed the reviews with one another and the Reviewing Editor has drafted this decision to help you prepare a revised submission.

Summary:

The manuscript reports an impressive and extensive analysis of mitotic recombination events and the first chromosome wide analysis of mitotic recombination in Mlh1 deficient *S. cerevisiae*.

The quality of the data, the results and interpretations provide an interesting set of information that could be of interest for a large audience. The authors have made significant effort to introduce the topic and the questions raised for specialists as well as non-specialists. However, as reviewers note, the analysis of the results would require much improvement to make it accessible.

The majority of events can be explained by current DSB repair models. However, additional features, such as discontinuous heteroduplex and/or mismatch repair activity, branch migration and template switching have to be implemented in order to explain other events. The conclusion could actually be presented more constructively by saying that this study analyzes properties of intermediates at distance from the initiating lesions and that the DSB repair model was not built on analysis. This analysis reveals properties that were not or partially addressed by other studies.

Essential revisions:

Several issues should be clarified when presenting the system and analyzing the data. These are essential and major modifications for publication in *eLife* such as to clarify the biological system being used and to allow the conclusions of this work to be accessible.

1) The nature of the initiation lesion. Comments should be made about processing of UV induced thymidine dimers: single strand nicks, double strand breaks.

2) The use of the Mlh1 mutation and consequences. One question linked to the use of Mlh1 is to evaluate and/or discuss whether mismatches could be involved despite the use of Mlh1 mutant. This mutant is still able to perform heteroduplex rejection. The authors should address this and compare with the context of an Msh2 mutant for instance.

3) Down regulation of HR in G1 and potential consequences on results should be discussed.

4) A developed summary section within the results should be presented to highlight the different threads of argument around the main points. For example, the strikingly high proportion of events involving breaks at the same site on two chromatids, ascribed to the replication of an unrepaired G1 break. The assay looks for crossovers and given that the sister chromatid is the preferred donor, G2 events will essentially be hidden. Similarly, downregulation of HR in G1 would potentially mean more G1 breaks would arrive in S unrepaired (would an EJ mutant show more of these events?

5) Figures: The data about events identified and analyzed is quite difficult to follow.

Information from supplementary figures should be merged: for instance, Figure S92 explains Figure S3.

These two figures should be combined (panel A of Figure S3 could actually be removed to save space and presented separately). In addition, the information from [Supplementary-material SD1-data]-Table S4, which describes by a sentence the nature of the event, should be provided in that merged figure also.

6) Tables: Besides the nature of events, it is difficult to get a whole picture of the relative contribution of the different events. Data is given in the text and in sup tables but not easy to access and decipher. Proposition can be: to present table(s) where the total number of each category of event are indicated. This should include the SCB/DSCB; selected/unselected; CO/NCO +/- CO/NCO; NCO/CO mechanism etc. A possibility is also to consider separately all the NCOs and all the COs and provide classes with their major characteristics.

7) Since SNPs scored are about 1kb apart, it is likely that much NCO heteroduplex (and possibly partner switches) have been missed. While this is acknowledged, the analysis would be stronger if an estimate of just how much has been missed (see Martini et al. for an example).

8) Subsection”Simple classes of CO and NCO chromatids*”* Coic et al. is cited in support of short-patch mismatch repair in budding yeast. I am not aware of another study, and since this is an important point, the validity of Coic et al.'s conclusions should be discussed in more detail.

---

## [Author Response]

*The manuscript reports an impressive and extensive analysis of mitotic recombination events and the first chromosome wide analysis of mitotic recombination in Mlh1 deficient S. cerevisiae.*

*The quality of the data, the results and interpretations provide an interesting set of information that could be of interest for a large audience. The authors have made significant effort to introduce the topic and the questions raised for specialists as well as non-specialists. However, as reviewers note, the analysis of the results would require much improvement to make it accessible.*

*The majority of events can be explained by current DSB repair models. However, additional features, such as discontinuous heteroduplex and/or mismatch repair activity, branch migration and template switching have to be implemented in order to explain other events. The conclusion could actually be presented more constructively by saying that this study analyzes properties of intermediates at distance from the initiating lesions and that the DSB repair model was not built on analysis. This analysis reveals properties that were not or partially addressed by other studies.*

In the Summary statement at the beginning of the decision letter, the reviewers acknowledge our efforts to explain a complex set of data, but suggest a large number of revisions to make the manuscript more accessible to general readers. The first point that they mention is that we should explain why our approach has advantages over previous studies of recombination. In the revised manuscript, we inserted the following paragraph at the end of the Introduction: “The model of recombination shown in Figure 1 is based primarily on studies of meiotic and mitotic recombination in yeast. […] The patterns of recombination observed in our study are likely to reflect a more realistic assessment of the complex events that occur during recombination. Similar conclusions have been reached by global analyses of meiotic recombination in yeast using methods similar to those employed in our study (Mancera *et al.* 2008; Martini *et al.* 2011).”

*Essential revisions:*

*Several issues should be clarified when presenting the system and analyzing the data. These are essential and major modifications for publication in eLife such as to clarify the biological system being used and to allow the conclusions of this work to be accessible.*

*1) The nature of the initiation lesion. Comments should be made about processing of UV induced thymidine dimers: single strand nicks, double strand breaks*

The reviewers wanted us to discuss the nature of the initiating DNA lesion introduced by UV. In particular, they wanted us to state whether the initiating DNA lesion is a double-stranded DNA break or a single-stranded nick. To address this comment, we added a paragraph to the beginning of the “Analysis of selected crossovers on chromosome V and unselected events induced by UV in YYy310” section of the Results:

“We found previously (Yin and Petes, 2013) that more than half of the recombination events detected in G_1_-synchronized diploids treated with high doses of UV had the pattern of recombination indicative of the repair of a G_1_-associated DSB (Figure 3).[…] Based on our analysis of UV-induced LOH events in wild-type and *rad14* (NER-defective) cells (Yin and Petes, 2015), single-broken chromatids are likely generated two different ways: by replication of a chromosome with an NER-generated gap, and by Mus81-dependent processing of DNA structures formed when a replication fork is blocked by an unprocessed UV-generated lesion”.

*2) The use of the Mlh1 mutation and consequences. One question linked to the use of Mlh1 is to evaluate and/or discuss whether mismatches could be involved despite the use of Mlh1 mutant. This mutant is still able to perform heteroduplex rejection. The authors should address this and compare with the context of an Msh2 mutant for instance.*

The reviewers requested that we rationalize the choice of using *mlh1* mutants rather than *msh2* mutants for our analysis. In particular, they suggested that some mismatches “could be involved” despite the use of the *mlh1* strains. They also point out that *mlh1* mutants could still perform heteroduplex rejection. To explain our choice of *mlh1*, we included the following sentences to the third paragraph of the Results section: “Both the Msh2 and Mlh1 proteins have central roles in the repair of mismatches resulting from bases misincorporated during DNA replication or formed within heteroduplexes during recombination. […] Although it is possible that some recombination intermediates could be lost in the *mlh1* strain by Msh2-dependent heteroduplex rejection, our analysis described below demonstrated that the level of mitotic recombination was not substantially affected by the *mlh1* mutation.”

*3) Down regulation of HR in G1 and potential consequences on results should be discussed.*

*4) A developed summary section within the results should be presented to highlight the different threads of argument around the main points. For example, the strikingly high proportion of events involving breaks at the same site on two chromatids, ascribed to the replication of an unrepaired G1 break. The assay looks for crossovers and given that the sister chromatid is the preferred donor, G2 events will essentially be hidden. Similarly, downregulation of HR in G1 would potentially mean more G1 breaks would arrive in S unrepaired (would an EJ mutant show more of these events?*

Points 3 and 4. The reviewers suggested a more extended discussion of the observation that many of recombination events are initiated in G1 rather than G2. They wanted this discussion to be within the Results section. The reviewers stated that we should point out that G2 events would likely be repaired by interaction with the sister chromatid and would be hidden. In addition, they mentioned that down-regulation of HR in G1 might be responsible for the high level of DSCB events. Finally, they suggested that elimination of NHEJ might result in even more DSCB events since these events would not be repaired in G1. In response to these comments, we included several new paragraphs into the “SCB and DSCB events are approximately equally frequent” section of the Results:

“Our conclusion that many of the observed recombination events, both spontaneous and UV-induced, reflect a G_1_-initiated DSB is surprising, but consistent with our previous studies (Lee *et al.* 2009; St. Charles and Petes, 2012; Yin and Petes, 2013). […] An alternative source of SCBs may be unexcised dimers that stall the replication fork, resulting in an S-phase-associated DSB (Yin and Petes, 2015).”

*5) Figures: The data about events identified and analyzed is quite difficult to follow.*

*Information from supplementary figures should be merged: for instance, Figure S92 explains Figure S3.*

*These two figures should be combined (panel A of Figure S3 could actually be removed to save space and presented separately). In addition, the information from [Supplementary-material SD1-data]- Table S4, which describes by a sentence the nature of the event, should be provided in that merged figure also.*

The reviewers recommend merging the supplementary figures such that the depiction of the segregation of the markers is on the same picture as the depiction of the proposed mechanisms. They also recommend including a sentence that describes the event as part of the figure. The figures were modified as requested.

*6) Tables: Besides the nature of events, it is difficult to get a whole picture of the relative contribution of the different events. Data is given in the text and in sup tables but not easy to access and decipher. Proposition can be: to present table(s) where the total number of each category of event are indicated. This should include the SCB/DSCB; selected/unselected; CO/NCO +/- CO/NCO; NCO/CO mechanism etc. A possibility is also to consider separately all the NCOs and all the COs and provide classes with their major characteristics.*

The reviewers requested that we include a summary table or tables that describe the classes of events and the numbers in each class. We added this table (Table 1) as requested. We also added Tables S6-S9 ([Supplementary-material SD1-data]) as additional summaries for event classes and tract lengths associated with each class of events. References to these tables in the main text are in subsection “*Simple classes of CO and NCO chromatids”* for Table 1, subsection “*SCB and DSCB events are approximately equally frequent”* for Table S6, subsection “*Lengths of conversion/heteroduplex tracts in UV-treated YYy310”* for Table S7, subsection “Effect of the *mlh1* mutation on the frequency of mitotic crossovers” for Table S8, and subsection “Analysis of the HS4 mitotic recombination hotspot” for Table S9.

*7) Since SNPs scored are about 1kb apart, it is likely that much NCO heteroduplex (and possibly partner switches) have been missed. While this is acknowledged, the analysis would be stronger if an estimate of just how much has been missed (see Martini et al., for an example).*

The reviewers point out that we may have missed detecting some NCO events and partner switches because the average spacing of the SNPs was about 1 kb. They suggested that we estimate the fraction of undetected events using the approach of Mancera et al. and Martini et al. This point is a good one. It should be noted, however, that the fraction of undetected events is likely to be less of a problem in our study than in meiotic studies because the median size of the mitotic NCO conversion tracts (5.4 kb) is considerably greater than the average inter-SNP distance. Nonetheless, we used the methods of Mancera et al. and Martini et al. to calculate what fraction of NCO events were missed. This analysis (described below) indicates that we detected about 80% of the NCO tracts. The sentences added to the manuscript were: “Since the average distance between SNPs, as assayed by the microarray, is about 1 kb, a fraction of short heteroduplex/conversion tracts will be undetectable (St. Charles et al. 2012; Zheng et al. 2016). We estimated the fraction of undetected events using the same procedure as Mancera et al. (2008) and Martini et al. (2011). Assuming that all crossovers are associated with heteroduplexes on both interacting chromatids, we determined what fraction of such chromatids had an observable heteroduplex or conversion ([Supplementary-material SD1-data]-Table S3). Of 33 crossovers, 23 had regions of heteroduplex on both recombined chromatids, 8 had only one region of heteroduplex on the two chromatids, and 2 had no detectable heteroduplex on either chromatid. Thus, we estimate that our methods detect about 80% (54/66) of the conversion events that are unassociated with crossovers. This calculation is dependent on the assumption that the lengths of the crossover-associated conversions are similar to those of the conversions that are unassociated with crossovers.

*8) Subsection” Simple classes of CO and NCO chromatids” Coic et al. is cited in support of short-patch mismatch repair in budding yeast. I am not aware of another study, and since this is an important point, the validity of Coic et al.'s conclusions should be discussed in more detail.*

The reviewers pointed out that there was only a single study (Coïc et al.) that argued for the existence of short-patch repair, and requested that we discuss the validity of the study in more detail. In the revised manuscript, we extended the discussion of the Coïc et al. paper as follows: “Coïc et al. (2000) reported the existence of a short-patch (<12 bp) Msh2-independent repair pathway in yeast. This pathway was hypothesized to explain a small number of recombination events in msh2 strains in which regions of unrepaired mismatches were interspersed with repaired mismatches. No genes in this pathway have been identified, although Coïc et al. (2000) argued that the nucleotide excision repair proteins were not involved.